



# Modeling nitrate from land-surface to wells' perforations under agricultural land: success, failure, and future scenarios in a Mediterranean case study.

Yehuda Levy[1], Roi H. Shapira[2], Benny Chefetz[3] and Daniel Kurtzman[4]

[1]Hydrology and Water Resources Program, The Hebrew University of Jerusalem, The Edmond J. Safra Campus - Givat Ram, 9190401 Jerusalem, Israel
[2]Mekorot, Israel National Water Company, Lincoln 9, 6713402 Tel-Aviv, Israel
[3]Department of Soil and Water Sciences, Faculty of Agriculture, Food and Environment, The Hebrew University of Jerusalem, 7610001 Rehovot, Israel.
[4]Institute of Soil, Water and Environmental Sciences, Volcani Center, Agricultural Research Organization, HaMaccabim Road 68, 7505101 Rishon LeZion, Israel

*Correspondence to*: Yehuda Levy (Yehuda.Levy1@mail.huji.ac.il)

**Abstract.** Contamination of groundwater resources by nitrate leaching under agricultural land is probably the most troublesome agriculture-related water contamination worldwide. Deep soil sampling (10 m) was used to calibrate vertical flow and nitrogen-transport numerical models of the unsaturated zone under different agricultural land uses. Vegetable fields (potato and strawberry) and deciduous orchards (persimmon) in the Sharon area overlying the coastal aquifer of Israel were examined. Average nitrate-nitrogen fluxes below vegetable fields were 210–290 kg ha$^{-1}$ yr$^{-1}$, and under deciduous orchards, 110–140 kg ha$^{-1}$ yr$^{-1}$. The output water and nitrate-nitrogen fluxes of the unsaturated zone models were used as input data for a three-dimensional flow and nitrate-transport model in the aquifer under an area of 13.3 km$^2$ of agricultural land. The area was subdivided into four agricultural land uses: vegetables, deciduous orchards, citrus orchards and non-cultivated. Fluxes of water and nitrate-nitrogen below citrus orchards were taken from a previous study in the area. The groundwater flow model was calibrated to well heads by changing the hydraulic conductivity. The nitrate-transport model, which was fed by the abovementioned models of the unsaturated zone, succeeded in reconstructing the average nitrate concentration in the wells. However, this transport model failed in calculating the high concentrations in the most contaminated wells and the large spatial variability of nitrate concentrations in the aquifer. To reconstruct the spatial variability and enable predictions, nitrate fluxes from the unsaturated zone were multiplied by local multipliers. This action was rationalized by the fact that the high concentrations in some wells cannot be explained by regular agricultural activity, and are probably due to malfunctions in the well area. Prediction of the nitrate concentration 40 years in the future with three nitrogen-fertilization scenarios showed that: (i) under the "business as usual" fertilization scenario, the nitrate concentration (as NO$_3^-$) will increase on average by 19 mg L$^{-1}$; (ii)under a scenario of 25 % reduction of nitrogen fertilization, the nitrate concentration in the aquifer will stabilize; (iii) with a 50 % reduction of nitrogen fertilization, the nitrate concentration will decrease on average by 18 mg L$^{-1}$.



## 1 Introduction

### 1.1 Groundwater contamination by nitrate under agricultural land

Since the development of the Haber–Bosch process in 1910, in which ammonia ($NH_3$) is cheaply produced from atmospheric nitrogen ($N_2$), mineral nitrogen has become the most important and common fertilizer in modern intensive agriculture. This process earned Fritz Haber the Nobel Prize for chemistry in 1918 and its significance was emphasized for many decades thereafter (e.g. "the most important invention of the 20th century" – Smil, 1999; Erisman et al., 2008). However, nitrogen fertilization is commonly applied in surplus and leaches below the roots, mainly as the conservative anion nitrate ($NO_3^-$),

which has strict limits under drinking-water standards worldwide. Thus nitrate has become the most common groundwater contamination caused by agricultural activity (Jalali, 2005; Vitousek et al., 2009; Burow et al., 2010; Kourakos et al., 2012; Yue et al., 2014; Wheeler et al., 2015; Wang et al., 2016). This contamination process occurs mainly below lighter soils and less under cultivated clays (Kurtzman et al., 2016). In Israel, more than half of the wells that have been disqualified as sources of drinking water were disqualified due to nitrate contamination (Israel Water Authority; IWA, 2015a).

### 1.2 The path from nitrogen fertilizer to nitrate in groundwater

Many studies have reported leaching ranges of 25–90 % of the nitrogen applied to agricultural fields in different crops and countries (Guimerá et al., 1995; McMahon and Woodside, 1997; Neilsen and Neilsen, 2002; Kraft and Stites, 2003; de Paz and Ramos, 2004; Ju et al., 2006; Zhao et al., 2011; Venterea et al., 2011). In Israel, Bar-Yosef et al. (1999) reported nitrate leaching of 55–65 % for different vegetables and field crops in a 35-year survey. More recently, Turkeltaub et al. (2015)

calculated leaching ratios in the range of 15–35 % under a modern greenhouse for intensive growing of vegetables.
Applications of nitrogen fertilizers of different species: nitrate, ammonium ($NH_4^+$) or organic nitrogen (e.g. urea, manure, compost) or a combination of these, are practiced. Most crops up-take only the mineral species (nitrate, ammonium). The nitrate and ammonium are up-taken by plant roots mostly in a mass transport process, which is limited by a crop-specific threshold concentration (Sorgona et al., 2006; Kurtzman et al., 2013). Some of the organic nitrogen in the soil is mineralized

to ammonium and in light soils, most of the ammonium is oxidized to nitrate (nitrification) in a relatively thin layer in the upper part of the soil column. Under anaerobic conditions, the nitrate is reduced to nitrogen gas via denitrification, which takes the nitrogen out of the system (Galloway et al., 2004). Nevertheless, denitrification is not a significant process in relatively aerated sandy soils and is frequently assumed to be negligible (Hanson et al., 2006; Doltra and Muñoz, 2010; Turkeltaub et al., 2015). Due to these processes, the nitrogen species that leaches down to the aquifer is mainly nitrate. In the

groundwater, nitrate is diluted and transported mostly as a conservative anion that is often extracted out of the system by pumping wells. Denitrification in aquifers is an important process in some cases (e.g., Thayalakumaran et al., 2015). Nevertheless, in the thick aquifer discussed here, dominated by sandy sediments and under Mediterranean climate, denitrification is negligible in the upper 95 % of the aquifer's depth (Kurtzman et al., 2012).



Nitrate contamination of the groundwater below agricultural land is often characterized by significant spatial distribution of
the nitrate concentrations in wells (Hu et al., 2005; Liu et al., 2005; Wheeler et al., 2015). This distribution may evolve from
the spatial distribution of the soil properties. Nevertheless, in an area with relatively uniform soil, it is most likely related to
variable land use (crops) and inconsistent agricultural practices (Almasri and Kaluarachchi, 2007; Bian et al., 2016).

Research of nitrate leaching from agricultural land can be divided into three scales and zones of interest. Agricultural aspects
of root uptake of nitrate and its seepage below the root zone have been studied quite extensively in the agricultural research
domain, where transient mechanistic models are often used for the analysis (e.g., Hanson et al., 2006; Doltra and Muñoz,
2010). The developing vadose-zone hydrology discipline looks at nitrate data and processes deeper in the unsaturated zone
as well (Kurtzman et al., 2013; Dahan et al., 2014). Regional assessments of groundwater contamination with nitrate make
use of varying degrees of simplification of vadose-zone processes (e.g., Mercado, 1976; de Paz and Ramos, 2004; Kourakos
et al., 2012).

The objective of this research was to quantitatively assess the nitrate throughout its course from fertilization on the field
surface through the flow processes in the root zone, down through the thick unsaturated zone, and in the aquifer toward the
pumping wells. We further aimed to restore the groundwater nitrate concentration by calculated fluxes from the unsaturated
zone and to explain the spatial distribution of the nitrate concentration in the groundwater by the spatial distribution of the
surface land use. Finally, we used the field- and regional-scale calibrated models for future assessment of aquifer
contamination under different fertilization scenarios.

## 2 Materials and methods

### 2.1 Research area: nitrate contamination in the Sharon area, Israel

The nitrate problem in groundwater in Israel is concentrated under intensively cultivated areas of Mediterranean red sandy-
loam (Hamra) soil overlying the coastal aquifer (IWA, 2015b; Kurtzman et al., 2016). Two main regions in which nitrate
contamination has been a concern for several decades are Rehovot–Rishon (Mercado, 1976) and the Sharon region
(Kurtzman et al., 2013). This research focuses on the Sharon area (Fig. 1). The Israeli coastal aquifer is an unconfined
aquifer, one of the most important freshwater sources in Israel for both agriculture and domestic consumption.

The climate is semiarid with annual precipitation of 550 mm mainly during the winter season from November to April. The
main land uses over the aquifer are agricultural and residential (cities, towns and villages).  The aquifer is in the Kurkar
group (Pleistocene) composed of sands, calcareous sandstone, and marine and continental silty and clay lenses. The aquifer
lies over the thick clays of the Saqiye group, which are conceptualized as an aquiclude (Gvirtzman, 2002). The unsaturated
zone thickness ranges from 3 to 80 m below ground surface.

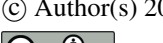



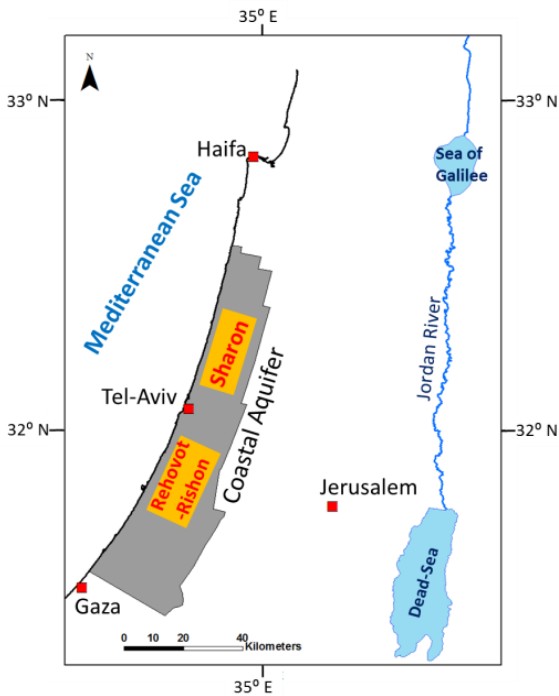

**Figure 1: Location map of the Israeli coastal aquifer and two areas (in red) with major nitrate contamination of the groundwater. This work presents a case study focusing on the Sharon area.**

This research concentrates on a 13.3km$^2$ agricultural area in the Sharon region. Nitrate concentration in wells in this research area have been increasing by an average 1 mg L$^{-1}$ yr$^{-1}$ for more than 40 years (Kurtzman et al., 2013). Although generally considered contaminated, significant spatial variability exists in the nitrate concentration in wells over short distances. Heavily contaminated wells can be at as little as 500 m from a non-contaminated well (Fig. 2).

The areal coefficient of variation in nitrate concentration is 38 % (Levy, 2015). This spatial variability indicates local contamination sources rather than regional contamination. It might evolve from crop type, fertilization masses or the agricultural practice in the fields at ground surface. Therefore, the research area was divided into four characteristic land uses: vegetables (40 % of area, large masses of nitrogen fertilization), citrus (33 % of area, also transpiring in the winter season), deciduous (14 % of area, large volumes of irrigation) and no crop (13 % of area) (land-use data from Survey of Israel maps, 2000).





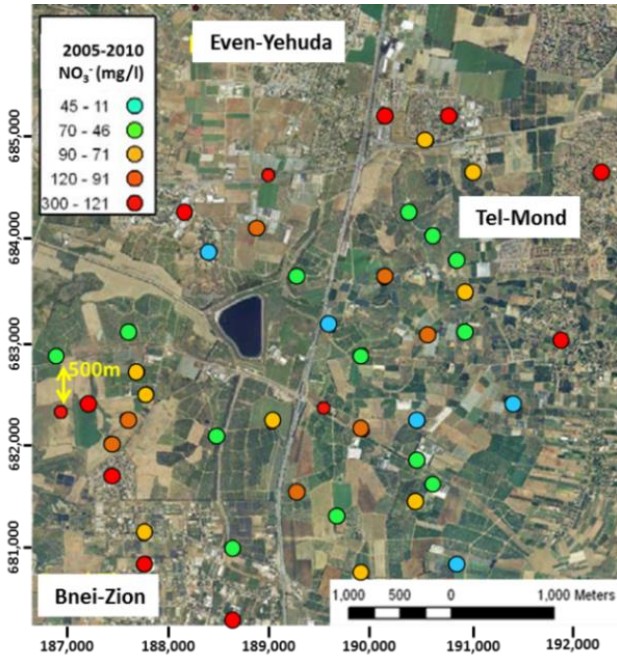

**Figure 2: The agricultural area selected for modeling and 5-year average nitrate concentration in wells. Note the high spatial variability. Nitrate concentration data are from the Israel Water Authority. Coordinates system: Israeli Transverse Mercator (ITM).**

## 2.2 Nitrate fluxes from the fields to the deep unsaturated zone

### 2.2.1 Fields, irrigation, fertilization and meteorological data

For the three aforementioned crop types, representative fields were selected for deep sampling in the Hamra soils of the Sharon region: potato and strawberry fields representing the vegetable land use; a persimmon plantation representing the deciduous crop, and data from an orange orchard reported in Kurtzman et al. (2013) representing citrus. In each field, data of irrigation and fertilization regimes (quantities and timing in daily resolutions) were collected from the farmers. Data on irrigation water quality (nitrate and chloride concentrations) were collected from the Israel Water Authority. The potato field was irrigated with an average irrigation depth of 480 mm yr$^{-1}$ and fertilized with 450 kg N ha$^{-1}$ yr$^{-1}$. The strawberry was irrigated to an average depth of 1000 mm yr$^{-1}$ and fertilized with 450 kg N ha$^{-1}$ yr$^{-1}$. Strawberries were grown under plastic tunnels and the field was completely covered with a plastic sheet during the winter, hence precipitation was not counted in the water balance for this field. The persimmon orchard was irrigated to an average depth of 850 mm yr$^{-1}$ and fertilized with 200 kg N ha$^{-1}$ yr$^{-1}$. The nitrogen forms of the applied fertilization were ammonium-nitrate solution in the irrigation water (persimmon and strawberry) and dry scattering of urea (potato). Nitrogen in the compost was accounted for in the strawberry and potato fields where this organic amendment was applied. The farmers in all representative fields reported that the same


crop was cultivated for at least 15 years before sampling (with minor exceptions for the potato field). Time series of daily precipitation and reference evapotranspiration (Penman–Monteith equation, Allen et al., 1998) for each field were collected from nearby automated meteorological stations operated by the Israel Ministry of Agriculture.

### 2.2.2 Field sampling and soil analysis

In each of the three fields (persimmon, strawberry and potato), three sampling coreholes were drilled using the direct push
technique, and a continuous core was obtained from 0–10 m depth (Fig. 3). The coreholes were drilled at a distance of 50–200 m from each other. Soil (and sediment) cores were cut into 30-cm segments. Drilling was done in June 2012. Core segments were sealed with caps and tape and kept in a cooler until reaching the laboratory, where the core segments were analyzed for the following variables: gravimetric water content (105 °C), bulk density (core dry mass per volume), gravimetric particle-size distribution (hydrometer method), chloride concentration of a 1:2 soil:water extract (with Sherwood
926 chloridometer), nitrate and ammonium concentrations in a 1 M KCl 1:5 soil:water solution extract (Kachurina et al., 2000, with Quickchem 8000 autoanalyzer, Lachat Instruments, Loveland, CO). The soil samples that were used for extraction were sieved to 2 mm after drying (40 °C for 3 days).

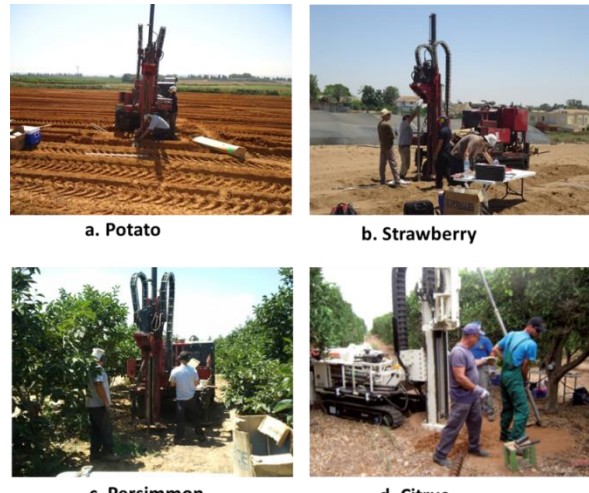

**Fig. 3. Direct push sampling of the unsaturated zone (0–10 m below ground surface) under the different agricultural land uses. (a–**
140 **c) Sampled in the current study. (d) Sampled for Kurtzman et al. (2013): the unsaturated model developed there was used here.**

### 2.2.3 Modeling water flow and nitrogen transport in the unsaturated zone

Steady-state approximations:
Average fluxes of water and nitrate-nitrogen toward the groundwater under the fields were calculated in a steady-state approximation with the chloride mass balance (Allison and Hughes, 1983; Scanlon et al., 2007):





$$R = \frac{(P \cdot Cl_p + I \cdot Cl_I) \int_{z=10m}^{z=2m} \theta(z)dz}{\int_{z=10m}^{z=2m} \theta(z) \cdot Cl_{PW}(z)dz} \ ,$$
(1)

where R [L T$^{-1}$] is the mean annual groundwater recharge flux, P [L T$^{-1}$] is the mean annual precipitation flux, I [L T$^{-1}$] is the mean annual irrigation application, Cl [M L$^{-3}$] is the steady-state approximation of the chloride concentration with subscripts P, I and PW referring to precipitation, irrigation water and unsaturated-zone pore water, respectively, and θ [L$^3$ L$^{-3}$] is the volumetric water content. The interval of integration for calculating deep unsaturated-zone averages was from z =

2 m (below the root zone) to z = 10 m depth (deepest available data). The steady-state approximation of nitrate flux to the groundwater was obtained by multiplying the water flux (R, Eq. 1) by the depth- and θ-weighted average of nitrate-nitrogen concentrations below the root zone:

$$F_{NO_3} = \frac{R \int_{z=10m}^{z=2m} \theta(z) \cdot NO_3 - N_{PW}(z)dz}{\int_{z=10m}^{z=2m} \theta(z)dz} \ ,$$
(2)

where $F_{NO3}$ [M L$^{-2}$ T$^{-1}$] is the mean annual flux of nitrate-nitrogen to the groundwater and NO$_3$-N$_{PW}$ [M L$^{-3}$] is the nitrate-

nitrogen concentration in the deep vadose zone pore water.

Transient models:

Transient vertical 1D numerical models of water flow and nitrogen transport were calibrated to data of one drill hole in each field: potato, strawberry and persimmon. The numerical code HYDRUS-1D was used for the calibration and simulations (Šimůnek et al., 2009). The 1D vertical Richards' equation with a root water-uptake sink was used for modeling flow in the

unsaturated zone:

$$\frac{\partial \theta(h)}{\partial t} = \frac{\partial}{\partial z}\left[K(h) \cdot \left(\frac{\partial h}{\partial z} + 1\right)\right] - S(h) \ ,$$
(3)

where t [T] is the time, z [L] is the vertical coordinate, h = h(z,t) [L] is the pressure head, θ(h) is the volumetric water content, K(h) [L T$^{-1}$] is the unsaturated hydraulic conductivity, and S(h) [T$^{-1}$] is a root water-uptake sink term which is non-zero in a transpiring root zone. The van Genuchten–Mualem model (Mualem, 1976; van Genuchten, 1980) was used for the

θ(h) and K(h) relationships of the different sediment layers, and Feddes et al.'s (1978) functions, fitted to each crop, were used for S(h) (Šimůnek et al., 2009).

One dimensional advection–dispersion equations representing chain reactions of the nitrogen system are presented in Eqs. (4–6). Only ammonium is accounted for in the solid phase. Sink/source terms for: root uptake of ammonium and nitrate, urea/compost mineralization, ammonium volatilization, ammonium nitrification and nitrate denitrification complete the

right-hand side of this equation system.

$$\frac{\partial \theta C_{Ur}}{\partial t} = \frac{\partial}{\partial z}\left(\theta D \frac{\partial C_{Ur}}{\partial z}\right) - \frac{\partial q C_{Ur}}{\partial z} - \mu_{min}\theta C_{Ur} \ ,$$
(4)

$$\frac{\partial \theta C_{NH4}}{\partial t} + \frac{\partial \rho K_d C_{NH4}}{\partial t} = \frac{\partial}{\partial z}\left(\theta D \frac{\partial C_{NH4}}{\partial z}\right) - \frac{\partial q C_{NH4}}{\partial z} - f_{NH4}SC_{NH4} + \mu_{min}\theta C_{Ur} - \mu_{nit}\theta C_{NH4} - \mu_{vol}\theta C_{NH4} \ ,$$ (5)





$$\frac{\partial \theta c_{NO3}}{\partial t} = \frac{\partial}{\partial z}\left(\theta D \frac{\partial c_{NO3}}{\partial z}\right) - \frac{\partial q c_{NO3}}{\partial z} - f_{NO3} S \cdot C_{NO3} + \mu_{nit}\theta C_{NH4} - \mu_{dnit}\theta C_{NO3} \;, \tag{6}$$

where $C_{Ur}$, $C_{NH4}$, and $C_{NO3}$ [M L$^{-3}$] are concentrations of the nitrogen species (urea, ammonium and nitrate, respectively) in the porewater solution, $\rho$ [M L$^{-3}$] is the soil's bulk density, $\theta$ [L$^3$ L$^{-3}$] is volumetric water content, D [L$^2$ T$^{-1}$] is the hydrodynamic dispersion coefficient, q [L T$^{-1}$] is the water flux, $f_{NH4}SC_{NH4}$ and $f_{NO3}SC_{NO3}$ [M T$^{-1}$ L$^{-3}$] are the root ammonium-nitrogen- and nitrate-nitrogen-uptake sinks, respectively, where $f_{NH4}$ and $f_{NO3}$ are user-defined functions relating solute uptake to the water uptake S and solute concentrations; $\mu_{min}$ [T$^{-1}$] is a first-order urea/compost mineralization rate (sink term in Eq. 4 and source term in Eq. 5), $\mu_{nit}$ [T$^{-1}$] is a first-order nitrification rate (sink term in Eq. 5 and source term in Eq. 6), $\mu_{vol}$ [T$^{-1}$] is a first-order ammonium-nitrogen volatilization rate, $\mu_{dnit}$ [T$^{-1}$] is a first-order denitrification rate and $k_d$ [L$^3$ M$^{-1}$] is the ammonium-nitrogen partition coefficient. Application of compost (strawberry) was treated with equations (4–6) as follows: farmers' reports of annual application of compost (m$^3$ ha$^{-1}$) were converted to mineralized nitrogen (Eq. 4) according to 15 % and 5 % nitrogen by mass mineralized in the first and second year after application, respectively (Eghball et al., 2002). A dry compost density of 600 kg m$^{-3}$ with 2 % of the dry mass consisting of nitrogen were used (Ben Hagai et al., 2011).

Fifty years (1962–2012) of daily precipitation, reference evapotranspiration (approximated from pan evaporation before 2002), irrigation water (with appropriate chloride and nitrate concentrations) and nitrogen fertilization were set as the upper boundary condition. A "Free Drainage" boundary (pressure gradient = 0) was used as the bottom boundary condition throughout. The calibration was aimed at fitting the measured profiles on the day of sampling, which was the last day of the 50 years of model runs.

Rosetta pedotransfer functions (Schaap et al., 2001) were used with particle-size distribution and bulk-density data to obtain initial values of the parameters of the hydraulic function $\theta(h)$ and K(h) for the model layers in the top 10 m (which were sampled and analyzed). These initial values were slightly changed during the calibration of the flow model in which the error between measured and modeled water contents was minimized. Dispersion coefficients of the soil/sediment layers were calibrated in the transport models with the unsaturated zone chloride observations. Nitrate-nitrogen data were used for calibrating the nitrate, mostly by changing the function of nitrate uptake, $f_{NO3}$ (Eq. 6). All calibrations were performed manually by trial-and-error runs.

To account for the actual unsaturated zone thickness in each cell of the groundwater model, the unsaturated models were extended/shortened to fit steady-state approximations of the actual unsaturated thickness (4–50 m below the ground, at 1m resolution, Fig. 4). This extension was also applied to the citrus orchard model from Kurtzman et al. (2013). Another model was constructed for water flow in the unsaturated zone below uncultivated areas using the hydraulic properties of the citrus orchard drill holes (this sampling point is at the center of the modeled area).



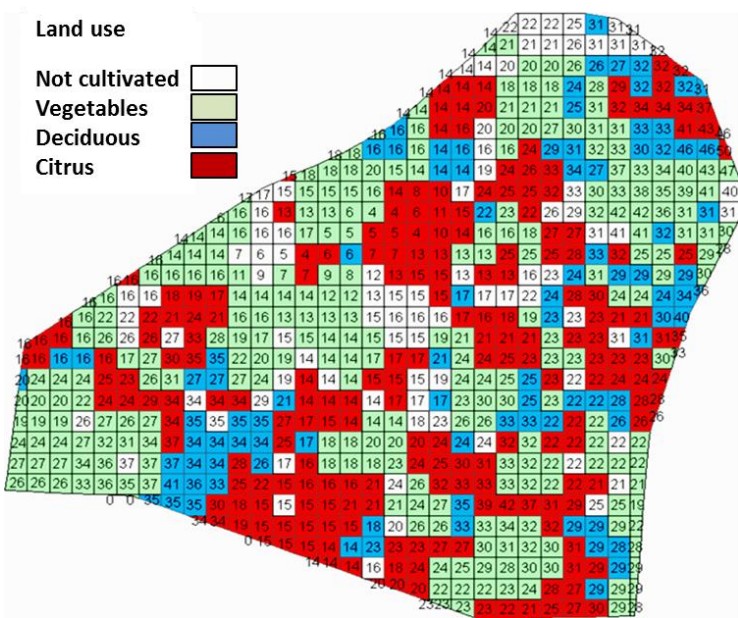

**Figure 4: Land use (color) and depth to water table in meters (number) for each grid cell of the modeled area.**

Thus, we created a "data library" of transient water and nitrate fluxes at the water table beneath the four land uses (posteriorly the potato model was used for the vegetable land use because the strawberry deep fluxes were similar and the potato field covered a greater area).

**2.3 Modeling of water flow and nitrate transport in the aquifer**

**2.3.1 Boundaries, data, spatial discretization, and simulation period**

A water flow and nitrate transport numerical model in the groundwater below the agricultural area in the Sharon region was developed. The model was constructed with GMS 8.2 software (AQUAVEO, 2012), the MODFLOW model for water flow (McDonald and Harbaugh, 1988) and the MT3D model for transport (Zheng, 1990). The model solves the water flow and advection–dispersion equations in the groundwater numerically (Eqs. 7 and 8):

$$S_S \cdot \frac{\partial h}{\partial t} = \frac{\partial}{\partial x}\left(K_{xx}\frac{\partial h}{\partial x}\right) + \frac{\partial}{\partial y}\left(K_{yy}\frac{\partial h}{\partial y}\right) + \frac{\partial}{\partial z}\left(K_{zz}\frac{\partial h}{\partial z}\right) + R - P \ , \tag{7}$$

$$\frac{\partial C}{\partial t} = \frac{\partial}{\partial x}\left(D_x\frac{\partial C}{\partial x}\right) - \frac{\partial(v_x C)}{\partial x} + \frac{\partial}{\partial y}\left(D_y\frac{\partial C}{\partial y}\right) - \frac{\partial(v_y C)}{\partial y} + \frac{\partial}{\partial z}\left(D_z\frac{\partial C}{\partial z}\right) - \frac{\partial(v_z C)}{\partial z} + \frac{R \cdot C_{duz}}{n} - \frac{P \cdot C}{n} \ , \tag{8}$$

where $S_s$ [L$^{-1}$] is the specific storage, h [L] is the hydraulic head, t [T] is the time, x,y,z [L] are the three-dimensional coordinates, $K_{xx}$, $K_{yy}$ and $K_{zz}$ [L T$^{-1}$] are the hydraulic conductivities along the x, y, z axes, P and R [T$^{-1}$] are volumetric fluxes per unit volume that represent sinks of water pumping in wells (P) and sources of water from recharge (R). C [M L$^{-3}$] is nitrate concentration in the aquifer, $D_x$, $D_y$ and $D_z$ [L$^2$ T$^{-1}$] are hydrodynamic dispersion coefficients, $v_x$, $v_y$ and $v_z$ [L T$^{-1}$]



are the velocities, n is porosity, and $C_{duz}$ [M L$^{-3}$] is the nitrate concentration in the deep unsaturated zone (in the recharge flux). The last term on the right in Eq. (8) is the nitrate sink due to pumping.

The modeled area was a polygon of 13.3 km$^2$ of agricultural land in the Sharon region of Israel. There has been no significant residential land use in this area in the last 60 years and all nitrate fluxes from the ground surface were assumed to be from agricultural sources. The boundary conditions were transient hydraulic heads and nitrate concentrations based on

data from wells near the model boundaries. Model calibration was based on transient measured data in wells inside the polygon (Fig. 5a). Time series of well heads and nitrate concentrations for the boundary conditions and calibration were obtained from the IWA.

The area was discretized to cells of 150:150 m. Vertically, the model is of 13 layers with thicknesses set according to the wells' perforations (Fig. 5b and 5c). Each cell in the top layer is fed with specific transient fluxes of water and nitrate from

230 the unsaturated zone, according to the unsaturated zone land-use model and its thickness (Fig. 4).

The groundwater model was run for 20 years (1992–2012). The input source/sink fluxes and boundary conditions were inserted into the model as monthly values (stress period = 1 month). By choosing this period, we ensured that the fluxes from the unsaturated zone (model runs start in 1962) represent the land use and not an artifact of initial conditions of the unsaturated zone models. Moreover, during the years 1992–2012, the average water level in the model regions was relatively

stable, supporting the steady-state approximation of the unsaturated zone thickness.

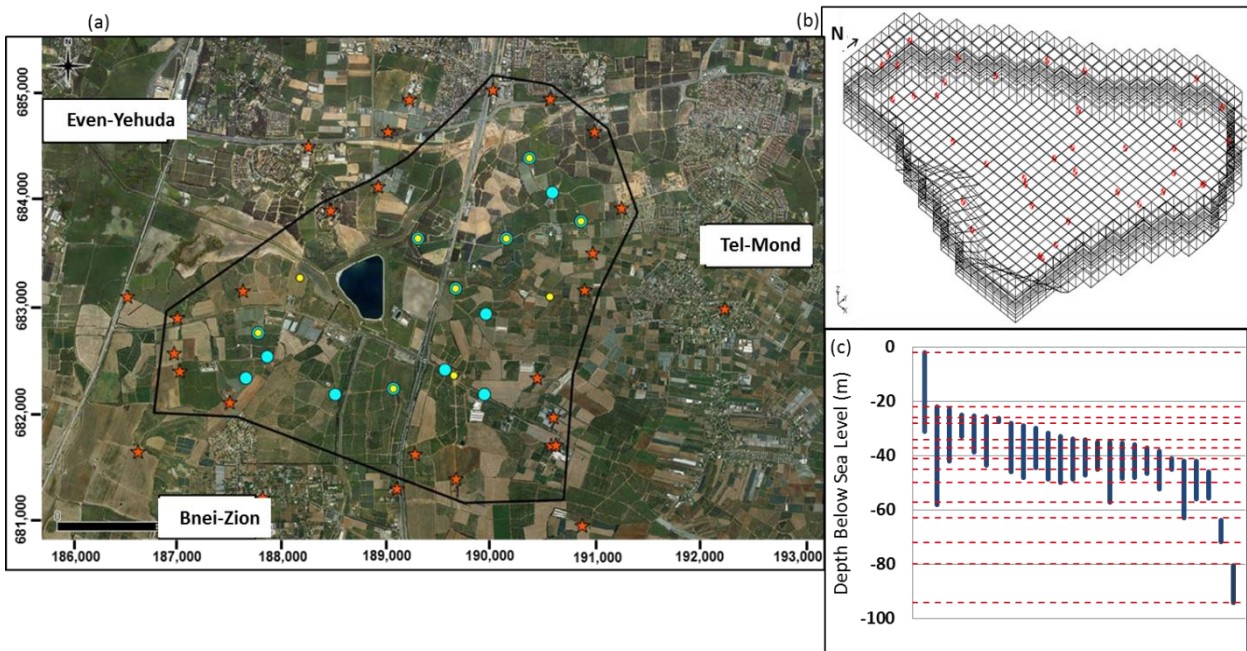

**Figure 5: (a) Groundwater model boundaries and wells: red stars – well data used for transient boundary conditions; yellow spots – well data used for calibration of the flow model; blue spots – well data used for calibration of nitrate transport model. (b) Depth of well screens (blue vertical bars) and model layers (red horizontal dashed lines). (c) 3D view of the model domain (finite**

**difference discretization) and wells (red).**





### 2.3.2 Groundwater model calibration

The water flow model was calibrated against measured water levels in the wells. The model was run with some zonation of horizontal hydraulic conductivity and a constant value of the storage coefficient until the mean absolute error (MAE) between measured and calculated water levels over the years was less than 0.5 m, and the mean error (bias) was close to zero. Recharge fluxes from the unsaturated zone model were strictly kept. In the first calibration stage of the nitrate transport model, dispersivity was fitted. Further steps in the calibration of this model were strongly related to the results and are elaborated upon in section 3.

### 2.3.3 Simulations of future nitrate contamination under various fertilization scenarios

An approximation based on the unsaturated modeling results reported by Kurtzman et al. (2013) was used to estimate the nitrate fluxes at the water table under different fertilization scenarios: a decrease of 25 % in the nitrogen fertilization mass results in a decrease of 50 % nitrate-nitrogen flux at the water table, whereas a reduction of 50 % in nitrogen fertilization results in a 72 % reduction in nitrate-nitrogen at the water table. Time series of nitrate-nitrogen fluxes at the water table were constructed using these ratios and the previously mentioned unsaturated flow and transport models (fitted to land uses and depth of the unsaturated zone). Three scenarios were tested: "business as usual", and 25 % and 50 % reduction in nitrogen application for the years 2012–2052. In these scenarios, it was assumed that land use would not change and that the precipitation and potential evapotranspiration would be similar to these data from 1972 to 2012. In the groundwater transport model, the transient nitrate-concentration boundary conditions were modified to account for similar reductions in nitrogen fertilization outside of the model domain.

## 3 Results

### 3.1 Unsaturated zone

### 3.1.1 Sediment data, and steady-state approximations of fluxes

Some spatial variability within the plot of each land use was observed, with one extremely different nitrate profile under the persimmon orchard (Fig. 6). Steady-state recharge and nitrate-nitrogen fluxes (Eqs. 1 and 2) were calculated for the data from each corehole. The spatial variability seen in the profiles (Fig. 6) was reflected by the variable deep fluxes within the plots (Table 1). Transient models were constructed for one corehole in each field. Nitrate-nitrogen fluxes under the strawberry and potato fields were relatively similar (~ 210 kg N ha$^{-1}$ yr$^{-1}$), hence the transient potato model that was calibrated to Profile C was used for all areas of vegetable land use. The transient model representing the deciduous land use



was Persimmon C (Table 1, Fig. 4). Hydraulically-significant lithologic data of the sediment profiles as gravimetric percentage of the clay texture (<0.002 mm) is displayed in table 2.

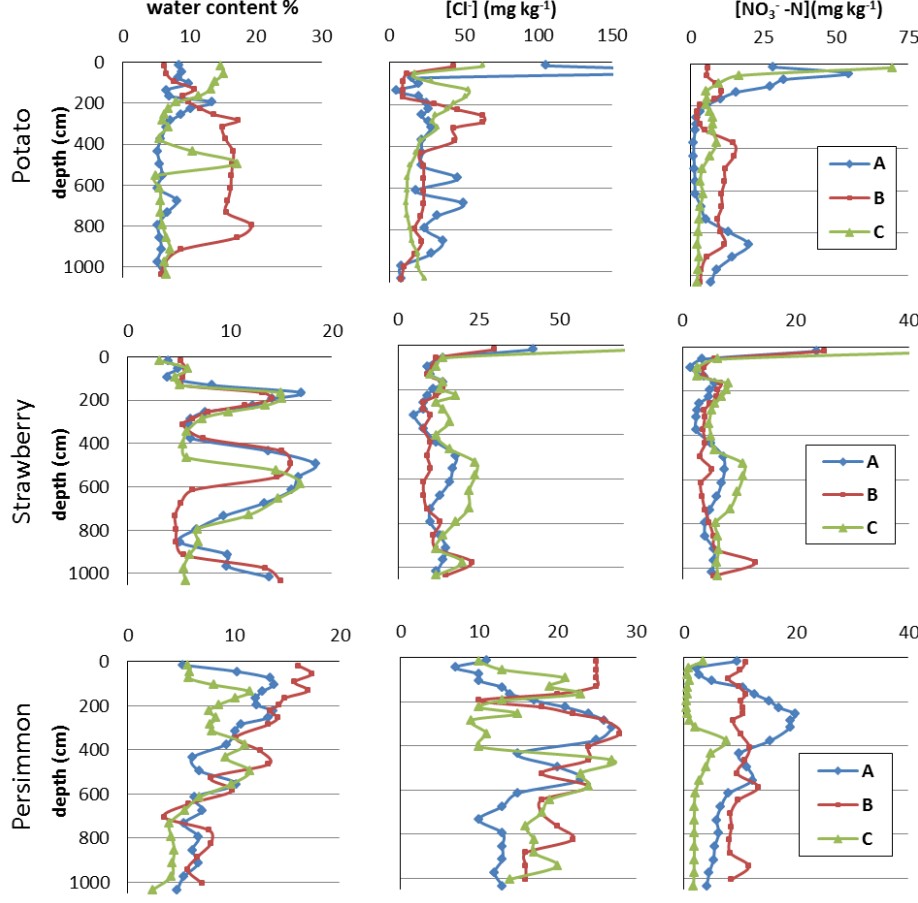

**Figure 6: Gravimetric water content and concentrations of chloride and nitrate-nitrogen in the sediment profiles. three sampling coreholes (A – blue, B – red, C – green) in each field (potato, strawberry and persimmon).**

**Table 1: Average deep (2–10 m) porewater concentrations and steady-state approximations of water and nitrate-nitrogen fluxes calculated for each profile.**

| | Potato | | | Strawberry | | | Persimmon | | |
|---|---|---|---|---|---|---|---|---|---|
| | **A** | **B** | **C** | **A** | **B** | **C** | **A** | **B** | **C** |
| Pore water mean chloride concentration (mg L$^{-1}$) | 421 | 192 | 266 | 198 | 179 | 188 | 234 | 232 | 263 |
| Pore water mean Nitrate-Nitrogen concentration (mg L$^{-1}$) | 96 | 63 | 63 | 47 | 53 | 76 | 130 | 25 | 38 |
| Water recharge flux (mm yr$^{-1}$) | 208 | 457 | 330 | 359 | 397 | 378 | 421 | 424 | 370 |
| Nitrate-Nitrogen flux (kg ha$^{-1}$ yr$^{-1}$) | 200 | 290 | 210 | 170 | 210 | 290 | 540 | 110 | 140 |





### 3.1.2 Transient unsaturated zone flow and nitrogen transport models

Table 2a–2c presents the hydraulic, transport and reaction model parameters that were calibrated to the observed unsaturated zone data. The partition coefficient for ammonium, $k_{d-NH4}$ = 3.5 L kg$^{-1}$ was used in all layers, and the first-order mineralization rate was set to $\mu_{min}$ = 0.56 day$^{-1}$ (Hanson et al., 2006). The relation of nitrate-nitrogen uptake to root-zone concentration and water uptake ($f_{NO3}SC_{NO3}$) was of the form used by Kurtzman et al. (2013) with limiting nitrate-nitrogen concentrations of 45 mg L$^{-1}$, 35 mg L$^{-1}$ and 20 mg L$^{-1}$ for potato, strawberry and persimmon, respectively. Limitation of the nitrogen reactions to the top layers of the soil was based on previous work in which nitrification potential was analyzed in orchard soils from this region (Kurtzman et al., 2013).

**Table 2. Measured clay content and parameters of the calibrated unsaturated zone flow and transport models under (a) potato field, (b) strawberry field, and (c) persimmon orchard. Note that in some layers hydraulic parameters were modified during calibrations (nd – no data).**

| (a) | | | Flow and transport parameters | | | | | | | Reaction parameters | | |
|---|---|---|---|---|---|---|---|---|---|---|---|---|
| | | | Water content | | $\alpha$ (cm$^{-1}$) | n | Saturate Hydraulic conductivity K (cm day$^{-1}$) | Bulk density $\rho$ (gr cm$^{-3}$) | Dispersivity (cm) | Volatilization (NH$_4$), Nitrification, Denitrification | | |
| Layer | Depth (m) | Clay (%) | Residual $\theta r$ | Saturation $\theta s$ | | | | | | $\mu_{vol}$ (day$^{-1}$) | $\mu_{nit}$ (day$^{-1}$) | $\mu_{dnit}$ (day$^{-1}$) |
| 1 | 0-0.15 | 19 | 0.068 | 0.415 | 0.025 | 1.6 | 68 | 1.45 | 1.5 | 0.05 | 0.2 | 0.005 |
| 2 | 0.15-0.3 | 19 | 0.068 | 0.415 | 0.025 | 1.6 | 68 | 1.45 | 1.5 | 0 | 0.2 | 0 |
| 3 | 0.3-0.45 | 19 | 0.068 | 0.415 | 0.025 | 1.6 | 68 | 1.45 | 1.5 | 0 | 0.05 | 0 |
| 4 | 0.45-1.5 | nd | 0.058 | 0.420 | 0.031 | 3.1 | 675 | 1.45 | 10 | 0 | 0 | 0 |
| 5 | 1.5-4 | 11 | 0.065 | 0.445 | 0.028 | 1.8 | 165 | 1.46 | 25 | 0 | 0 | 0 |
| 6 | 4-5.2 | 4 | 0.057 | 0.409 | 0.031 | 3.3 | 775 | 1.43 | 12 | 0 | 0 | 0 |
| 7 | 5.2-9.4 | 5 | 0.057 | 0.406 | 0.031 | 3.3 | 766 | 1.6 | 38 | 0 | 0 | 0 |
| 8 | 9.4-10.15 | 2 | 0.068 | 0.415 | 0.025 | 1.6 | 68 | 1.57 | 9 | 0 | 0 | 0 |
| 9 | 10.15-10.3 | nd | 0.065 | 0.445 | 0.028 | 1.8 | 165 | 1.46 | 25 | 0 | 0 | 0 |



| (b) | | | Flow and transport parameters | | | | | | | Reaction parameters | | |
|---|---|---|---|---|---|---|---|---|---|---|---|---|
| | | | Water content | | $\alpha$ (cm$^{-1}$) | n | Saturate hydraulic conductivity K (cm day$^{-1}$) | Bulk density $\rho$ (gr cm$^{-3}$) | Disper-sivity (cm) | Volatilization (NH$_4$), Nitrification, Denitrification | | |
| Layer | Depth (m) | Clay (%) | Residual $\theta r$ | Saturation $\theta s$ | | | | | | $\mu_{vol}$ (day$^{-1}$) | $\mu_{nit}$ (day$^{-1}$) | $\mu_{dnit}$ (day$^{-1}$) |
| 1 | 0 - 0.15 | 3 | 0.053 | 0.401 | 0.033 | 3.2 | 709 | 1.46 | 1.5 | 0.05 | 0.18 | 0.001 |
| 2 | 0.15 - 0.3 | 3 | 0.053 | 0.401 | 0.033 | 3.2 | 709 | 1.46 | 1.5 | 0 | 0.18 | 0.005 |
| 3 | 0.3 - 0.45 | 3 | 0.053 | 0.401 | 0.033 | 3.2 | 709 | 1.46 | 1.5 | 0 | 0.005 | 0 |
| 4 | 0.45 - 1.5 | 3 | 0.053 | 0.401 | 0.033 | 3.2 | 709 | 1.46 | 10.5 | 0 | 0 | 0 |
| 5 | 1.5 – 2.9 | 23 | 0.068 | 0.388 | 0.024 | 1.4 | 34 | 1.56 | 14 | 0 | 0 | 0 |
| 6 | 2.9 - 4.95 | 3 | 0.053 | 0.405 | 0.032 | 3.4 | 788 | 1.44 | 2 | 0 | 0 | 0 |
| 7 | 4.95 - 6.15 | 18 | 0.065 | 0.408 | 0.026 | 1.6 | 61 | 1.49 | 12 | 0 | 0 | 0 |
| 8 | 6.15 - 7 | 18 | 0.075 | 0.489 | 0.026 | 1.4 | 98 | 1.23 | 8.5 | 0 | 0 | 0 |
| 9 | 7 - 7.65 | 18 | 0.059 | 0.358 | 0.028 | 1.4 | 31 | 1.65 | 6.5 | 0 | 0 | 0 |
| 10 | 7.65 - 10.3 | 4 | 0.054 | 0.392 | 0.031 | 3.4 | 767 | 1.5 | 25 | 0 | 0 | 0 |

| (c) | | | Flow and transport parameters | | | | | | | Reaction parameters | | |
|---|---|---|---|---|---|---|---|---|---|---|---|---|
| | | | Water content | | $\alpha$ (cm$^{-1}$) | n | Saturate hydraulic conductivity K (cm day$^{-1}$) | Bulk density $\rho$ (gr cm$^{-3}$) | Disper-sivity (cm) | Volatilization (NH$_4$), Nitrification, Denitrification) | | |
| Layer | Depth (m) | Clay (%) | Residual $\theta r$ | Saturation $\theta s$ | | | | | | $\mu_{vol}$ (day$^{-1}$) | $\mu_{nit}$ (day$^{-1}$) | $\mu_{dnit}$ (day$^{-1}$) |
| 1 | 0-0.15 | 12 | 0.06 | 0.404 | 0.028 | 2 | 159 | 1.48 | 1.5 | 0.08 | 0.1 | 0.0025 |
| 2 | 0.15-0.3 | 12 | 0.06 | 0.404 | 0.028 | 2 | 159 | 1.48 | 1.5 | 0 | 0.01 | 0.001 |
| 3 | 0.3-0.45 | 12 | 0.06 | 0.404 | 0.028 | 2 | 159 | 1.48 | 7 | 0 | 0 | 0 |
| 4 | 0.45-1.2 | 12 | 0.06 | 0.404 | 0.028 | 2 | 159 | 1.48 | 20 | 0 | 0 | 0 |
| 5 | 1.2-2.1 | 15 | 0.059 | 0.367 | 0.028 | 1.6 | 60 | 1.61 | 9 | 0 | 0 | 0 |
| 6 | 2.1-3.45 | 12 | 0.056 | 0.364 | 0.030 | 1.9 | 114 | 1.61 | 13.5 | 0 | 0 | 0 |
| 7 | 3.45-5.9 | 11 | 0.055 | 0.353 | 0.030 | 1.7 | 62 | 1.65 | 24 | 0 | 0 | 0 |
| 8 | 5.9-7.05 | 4 | 0.057 | 0.392 | 0.030 | 3.1 | 599 | 1.49 | 12 | 0 | 0 | 0 |
| 9 | 7.05-10.3 | 2 | 0.053 | 0.353 | 0.030 | 4.5 | 1357 | 1.5 | 32 | 0 | 0 | 0 |





Water and nitrogen balances resulting from the calibrated models showed significant recharge and deep nitrate-nitrogen

leaching (40–55 % of total nitrogen input) under the investigated agricultural land (Table 3a and 3b). The yearly average (for

2002–2012) water and nitrate-nitrogen fluxes toward the water table calculated by the numerical models and those calculated

by the steady-state approximation (chloride mass balance) matched well (Table 3a and 3b). The maximal difference between

the two methods was 24 mm yr$^{-1}$ (6.5 %) and 20 kg ha$^{-1}$ yr$^{-1}$ (7 %) for the water and nitrate-nitrogen fluxes, respectively. The

average flux of nitrate-nitrogen toward the water table in citrus orchards in this area  was found to be 30 % of the total

nitrogen input (Kurtzman et al. 2013), lower than the leaching fraction under the vegetable and deciduous areas investigated

here.

**Table 3. Annual average (a) water and (b) nitrogen balance calculated by the unsaturated transient flow and transport models for 2002–2012, and comparison of deep fluxes to steady-state approximations.**

| (a) | | Potato | Strawberry | Persimmon |
|---|---|---|---|---|
| Average water input (mm yr$^{-1}$) | Irrigation | 463 | 1050 | 822 |
| | Rain | 607 | 0 | 538 |
| Average water output (mm yr$^{-1}$) | Root Uptake | 467 | 367 | 639 |
| | Evaporation | 276 | 335 | 352 |
| | Recharge | 323 | 354 | 366 |
| Recharge by CMB (mm yr$^{-1}$; wells C in Table 1) | | 330 | 378 | 370 |

| (b) | | Potato | Strawberry | Persimmon |
|---|---|---|---|---|
| Average nitrogen input (Kg Ha$^{-1}$ yr$^{-1}$) | Fertilization | 450 | Mineral-350 Organic-100 | 200 |
| | Nitrate-nitrogen in irrigation water | 50 | 100 | 90 |
| Average nitrogen output (Kg Ha$^{-1}$ yr$^{-1}$) | Ammonia-volatilization | 65 | 35 | 25 |
| | Denitrification | 65 | 75 | 35 |
| | Root ammonium-nitrogen uptake | 20 | 35 | 20 |
| | Root nitrate-nitrogen uptake | 165 | 125 | 110 |
| | **Nitrate-nitrogen flux toward groundwater** | **200** | **310** | **130** |
| Nitrate-nitrogen flux toward groundwater by chloride mass balance (Kg Ha$^{-1}$ yr$^{-1}$; wells C in Table 1) | | 210 | 290 | 140 |
| **Nitrate-Nitrogen leaching percentage** | | **40%** | **55%** | **45%** |



**300**    **3.2 Groundwater model**

**3.2.1 Model calibration**

The flow model was calibrated by assigning different horizontal hydraulic conductivities, in the range of $K_{xx} = K_{yy} = 4.5 – 30$ m d$^{-1}$, to five subregions, where the higher values are in the western part of the modeled area. These hydraulic conductivity values are similar to previous studies in the Sharon region of the Israel coastal aquifer (Bachmat et al., 2003; Lutsky and

**305**    Shalev, 2010). The calibrated anisotropy was $K_{xx} \ K_{zz}^{-1} = 5$ and the specific yield was $Sy = 0.12$.

The goodness-of-fit parameters between calculated and observed heads were the MAE and the mean error (the bias), calculated for each observation well (Table 4) and for all observations. The improvement in the calibration ceased when the target weighted-average MAE < 0.5 m and bias < 0.1 for all observations were met (Table 4, Fig. 7).

**Table 4: Goodness of fit of the calibrated flow model (calculated–observed). MAE – mean absolute error; bias – mean error.**

| Well Name | # of observations | MAE (m) | Bias (m) |
|---|---|---|---|
| Tel Mond  Ziv A | 8 | 0.31 | 0.15 |
| Tel Mond 8 | 6 | 0.40 | 0.21 |
| Herut 41/3 | 20 | 0.48 | 0.29 |
| Tel Mond 13 | 9 | 0.25 | <0.01 |
| Bnei Dror D | 1 | 0.31 | -0.31 |
| Tel Izhak C | 18 | 0.34 | -0.31 |
| Tel Izhak 41/2 | 20 | 0.61 | 0.49 |
| Gan Efraim 3 | 27 | 0.72 | -0.56 |
| Gan Efraim 2 | 8 | 0.19 | 0.10 |
| Gan Shlomo Berman-Cohen | 6 | 0.45 | 0.45 |
| Total observations and weighted-average errors | 123 | 0.48 | <0.01 |

**310**





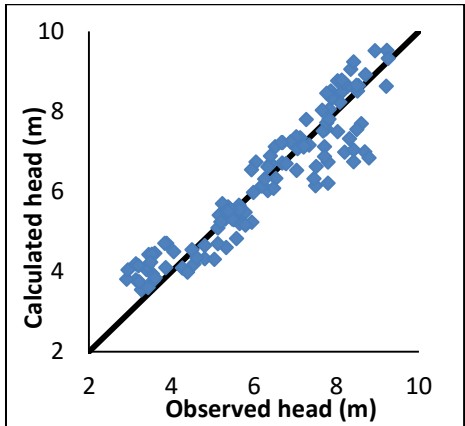

**Figure 7: Calibrated flow model's calculated vs. observed heads in meters above mean sea level. Black line: calculated = observed.**

The nitrate transport model was calibrated by changing the dispersivity value, starting with a value in line with Neuman's (1990) formula. The transport parameters used in the calibrated model were: dispersivity = 500 m, ratio between longitudinal and transverse dispersivities = 10 and effective porosity = 0.12. This first stage of the calibration resulted in a good fit between observed and modeled nitrate concentrations over the entire modeled area (i.e., spatially weighted average with weights for each well calculated by the Thiessen polygon method; Thiessen, 1911). However, the model showed poor fits between observed and calculated nitrate concentrations at each well separately (Table 5, Fig. 8a). The model reconstructed well the entire mass of nitrate in the aquifer but it failed to describe the nitrate's spatial variability (bottom two lines in Table 5a vs. observed, averages and standard deviations). To test whether the nitrate inputs from the unsaturated-zone model are significant in comparison to nitrate flowing from the boundaries (variable-concentration boundary condition), the model was run with 0 nitrate flux from the unsaturated zone. The overall average nitrate concentration was 0.66 of the observed concentration (bottom two lines in Table 5b vs. observed, averages and standard deviations). These results led to the understanding that although the unsaturated model produces good values for overall nitrate flux, the contaminated wells cannot be modeled with fluxes resulting from "normal" agricultural practice.

Simulations showed that observed nitrate concentrations above 100 mg L$^{-1}$ cannot be simulated with the nitrate fluxes produced by the calibrated unsaturated zone model (Table 3b). Multiplication of fluxes by up to a factor of 10 was needed to produce high concentrations in the wells. On the other hand, we had to maintain the overall flux of nitrate over the entire model domain. Therefore, in the second stage of the calibration, nitrate fluxes that were calculated by the unsaturated zone model were multiplied by factors as follows: 1 % of the area – factor of 10 (near the most contaminated wells); 3 % of the area – factor of 5; 4 % – factor of 2.8; 55 % – factor of 1; 19 % – factor of 0.6 and in 18 % of the area, the fluxes were multiplied by a factor of 0.1. The reasoning for these extreme fluxes in small areas surrounding some wells will be discussed later (in section 4). These local multipliers resulted in a reasonable fit between observed and modeled nitrate concentrations for both each well separately and the overall nitrate average and standard deviation (bottom two lines in Table 5c vs. observed).





**Table 5: Observed vs. calculated nitrate concentrations during the calibration process. (a) After the first calibration stage (parameter fit). (b) A test model with 0 nitrate flux from the unsaturated zone. (c) After the second calibration stage (local multipliers). Avg. – average; MAE – mean absolute error; bias – mean error.**

| Well Name | # of observations | Observed Avg. Value (mg L$^{-1}$) | (a) Model after 1st calibration | | | (b) Model without nitrate influx from unsaturated zone | | | (c) Model after 2nd calibration | | |
|---|---|---|---|---|---|---|---|---|---|---|---|
| | | | MAE (mg L$^{-1}$) | Bias (mg L$^{-1}$) | Calculated Avg. Value (mg L$^{-1}$) | MAE (mg L$^{-1}$) | Bias (mg L$^{-1}$) | Calculated Avg. Value (mg L$^{-1}$) | MAE (mg L$^{-1}$) | Bias (mg L$^{-1}$) | Calculated Avg. Value (mg L$^{-1}$) |
| Bnei Dror D | 14 | 20 | 48 | -48 | 68 | 20.8 | -20.1 | 40.1 | 8 | 2 | 18 |
| Tel Mond 5 | 10 | 51 | 13 | -12 | 64 | 6.8 | 6.6 | 44.5 | 4 | -3 | 54 |
| Tel Mond 8 | 31 | 53 | 15 | -2 | 55 | 14.1 | 10.2 | 42.7 | 14 | 1 | 52 |
| Herut 6 | 24 | 54 | 15 | -15 | 69 | 12.2 | 12.2 | 41.9 | 10 | -10 | 64 |
| Tel Mond Ziv A | 9 | 59 | 14 | -14 | 73 | 4.1 | 3.6 | 55.7 | 5 | -4 | 63 |
| Tel Izhak C | 13 | 61 | 15 | -13 | 73 | 17.5 | 17.5 | 43.0 | 8 | 7 | 53 |
| Gan Efraim 4 | 13 | 65 | 13 | -10 | 75 | 13.0 | 12.0 | 53.1 | 11 | -3 | 68 |
| Tel Mond 13 | 17 | 66 | 9 | 4 | 63 | 24.9 | 24.3 | 42.2 | 11 | -1 | 67 |
| Gan Shlomo Man | 10 | 70 | 11 | -6 | 76 | 19.1 | 16.0 | 54.0 | 13 | -9 | 79 |
| Gan Shlomo Berman | 15 | 75 | 10 | -0.3 | 75 | 22.4 | 22.4 | 52.6 | 10 | -2 | 77 |
| Gan Efraim 2 | 13 | 87 | 12 | 10 | 77 | 40.4 | 40.4 | 46.6 | 11 | -4 | 91 |
| Gan Efraim Lapter | 14 | 101 | 30 | 30 | 70 | 54.4 | 54.4 | 46.0 | 12 | 3 | 97 |
| Gan Shlomo A | 11 | 115 | 26 | 26 | 90 | 38.5 | 38.5 | 76.7 | 15 | 8 | 107 |
| Tel mond 3A | 14 | 130 | 59 | 59 | 72 | 86.7 | 86.7 | 43.7 | 19 | 12 | 119 |
| **All Wells** | 208 | **73** | 20 | 0.6 | **72** | 27 | 24.5 | **48** | 10 | 0.3 | **72** |
| **Standard Deviation** | | **27** | | | **8** | | | **9** | | | **25** |

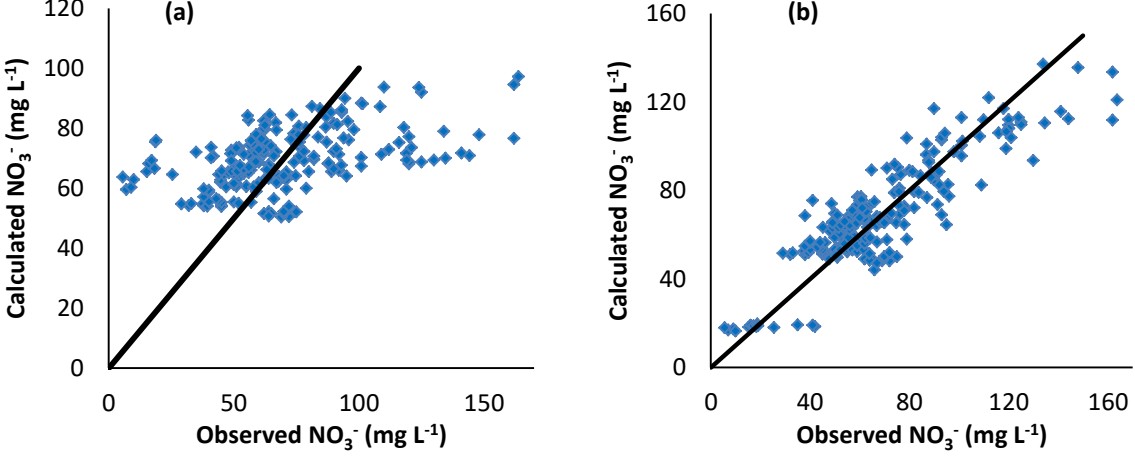

**Figure 8: Calculated vs. observed nitrate concentrations. (a) After the first calibration stage (parameter fit). (b) After the second calibration stage (local multipliers). Black line is calculated = observed.**





### 3.2.2 Simulations of three fertilization scenarios 40 years into the future

The calibrated model was run to 40 years in the future (2012–2052) under three scenarios: (i) "business as usual"; (ii) application of 75 % of the currently applied nitrogen fertilization; (iii) application of 50 % of the currently applied nitrogen

fertilization. The simulation results showed that (i) the average concentration in all wells in the simulated area will continue to increase in the "business as usual" scenario, reaching 106 mg L$^{-1}$ in 2052 (vs. 87 mg L$^{-1}$ in 2012); (ii) reducing the fertilization to 75 % will approximately maintain the present concentrations; (iii) reducing the fertilization to 50 % will lead to a trend of declining nitrate concentration to less than 70 mg L$^{-1}$ (Israel's drinking water standard for nitrate) on average for all wells in the modeled area (Fig. 9 and Table 6).

**Table 6: Current (2012) observed nitrate concentrations and those simulated for the year 2052 for three nitrogen-fertilization scenarios: 100 %, 75 % and 50 % of the current application used by farmers. In red are concentrations below the Israeli drinking water standard for nitrate.**

| Well Name | Observed (2012, mg L$^{-1}$) | Simulated concentrations at 2052 (mg L$^{-1}$) for fertilization scenario | | |
|---|---|---|---|---|
| | | 100 % | 75 % | 50 % |
| Bnei Dror D | 16 | 27 | 21 | 19 |
| Tel Mond 5 | 60 | 77 | 64 | 57 |
| Tel Mond 8 | 60 | 82 | 68 | 61 |
| Herut 6 | 69 | 79 | 67 | 60 |
| Tel Mond  Ziv A | 70 | 83 | 68 | 59 |
| Tel Izhak C | 73 | 96 | 73 | 62 |
| Gan Efraim 4 | 79 | 101 | 77 | 66 |
| Tel Mond 13 | 78 | 99 | 80 | 71 |
| Gan Shlomo Man | 88 | 107 | 84 | 73 |
| Gan Shlomo Berman-Cohen | 90 | 109 | 86 | 75 |
| Gan Efraim 2 | 106 | 138 | 101 | 84 |
| Gan Efraim Lapter | 122 | 139 | 98 | 78 |
| Gan Shlomo A | 128 | 130 | 103 | 89 |
| Gan Efraim 3 | 129 | 157 | 108 | 86 |
| Tel mond 3A | 134 | 164 | 112 | 88 |
| **Average** | **87** | **106** | **81** | **69** |
| **Standard Deviation** | 26.7 | 29.1 | 16.5 | 11.5 |



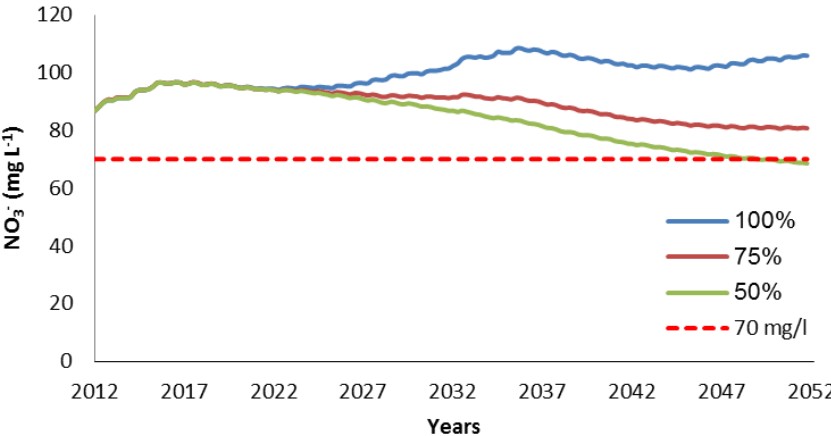

**Figure 9: Simulations of future average nitrate concentrations in wells under three nitrogen-fertilization scenarios: 100 %, 75 % and 50 % of the current application used by farmers. 70 mg L$^{-1}$ – Israel's drinking water standard for nitrate.**

## 4 Discussion

Our results showed successful evaluation of the total mass of nitrate in the aquifer using data of agricultural practice and deep unsaturated-zone samples to calibrate flow and transport models of the unsaturated zone, which feed the aquifer. Nevertheless, this straightforward model failed to produce the observed spatial variability of nitrate concentrations in wells, which required a random non-mechanistic modeling approach.

Successful delivery of the total volumes of water and nitrate mass to the 13.3 km$^2$ aquifer under agricultural land was achieved despite the following coarse assumptions: only four types of land use (three crops); steady crops for 50 years; homogeneity of agricultural practices and similar profiles of porous medium within each crop. These assumptions neglect small-scale variability, yet work for the regional scale totals for the following reasons: the farmers generally follow irrigation and fertilization recommendations made by extension services; about half of the land is covered by orchards for which applications of water and fertilizer have been steady for decades; on a regional scale, if the soil properties are generally similar, the details of the different profiles of the deep unsaturated zone have only a minor effect.

Failure to reproduce the spatial variability of nitrate concentrations lay mainly in predicting the extreme concentrations in some wells. These nitrate concentrations cannot be explained by any rational agricultural practice, and are a result of random failure of even fertilizer distribution in the field. It should be acknowledged that water wells are often at the "logistic center" of the agricultural field, and organic and mineral fertilizers are stocked nearby; temporal leakage can cause high concentrations in the well for years afterwards. Furthermore, the immediate area of the well is susceptible to preferential flow paths due to incidental ponding (Gurdak et al., 2008) and/or shortcuts through the annulus of the boreholes. This is especially common in old private boreholes that are used mainly for irrigation, which are common in the investigated area.



Heterogeneity of the porous medium may cause extremely high nitrate fluxes and may be a reason for contaminated wells. Of the nine deep profiles reported here, only one showed extreme nitrate concentrations and calculated nitrate fluxes that were 4- to 5-fold higher than in the other profiles extracted from the same orchard (Persimmon A, Table 1). Therefore, the non-physical multiplications used to calibrate the nitrate transport model were ultimately justified. Moreover, these multiplications were essential for simulating future scenarios (Fig. 9, Table 6).

In the case of the Israeli coastal aquifer, we are fortunate enough to be able to perform a post-audit analysis of nitrate-level predictions made 40 years ago in another part of the aquifer (Rehovot-Rishon region, Fig. 1). This region of the aquifer was overlain mainly by agricultural land (in 1950–1970), with similar sandy-loam (Hamra) soils (Mercado, 1976). The latter work predicted a continual increase in nitrate concentration in the groundwater below this area, from 50 mg L$^{-1}$ in 1970 to a range of 120–180 mg L$^{-1}$ in 2015 (Fig. 10). The observed average concentration in this area in 2014 was 90 mg L$^{-1}$ (Israel

Water Authority data). This is indeed an increase, but not the expected one. On the other hand, this increase of 40 mg L$^{-1}$ over 45 years is similar to the nitrate concentration increase in the Sharon area (Kurtzman et al., 2013).

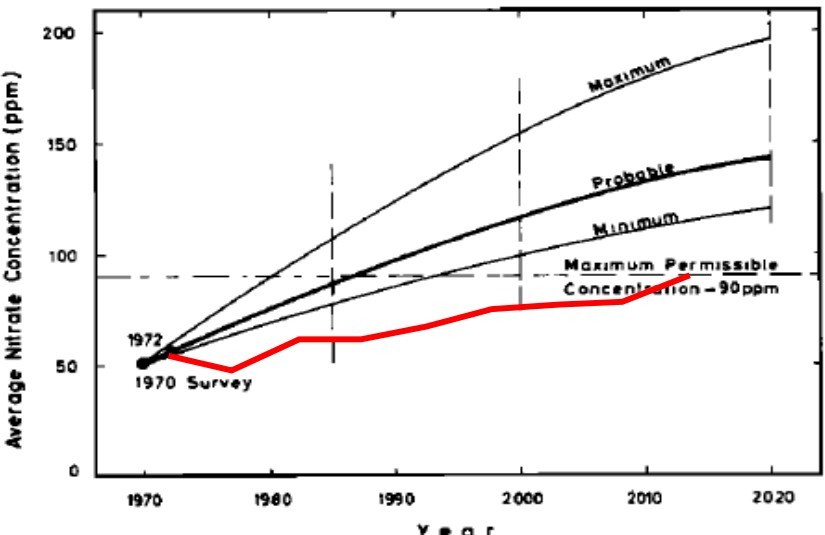

**Figure 10: Post-audit of average nitrate concentration predicted in 1976 for another part of the Israeli coastal aquifer. All black lines and writing are original predictions from Mercado (1976). Red line is the historical average nitrate concentrations from the**
390 **wells in that area that produced since 1970 (no new wells, data were obtained from the Israel Water Authority. Maximum permissible concentration of nitrate was reduced from 90 mg L$^{-1}$ to 70 mg L$^{-1}$ in 2001).**

The main reason for the overshoot of Mercado's (1976) prediction is probably the very significant reduction in agricultural land due to urbanization in this area in the last 5 decades. Most of this urbanization are agricultural towns which became modern cities with tight sewage systems, where practically all the wastewater is piped to treatment plants. In the current
work, the predictions were also made assuming steady agricultural land use with no urbanization processes that might lead to a similar overshoot in nitrate concentration predictions.





## 5 Summary and Conclusions

Groundwater under irrigated agricultural land over light soils commonly suffers from nitrate contamination. Nevertheless, significant spatial variability in nitrate concentrations in these parts of the aquifer exist, suggesting that it is caused by variability in nitrate fluxes from the unsaturated zone. An agricultural area (13.3 km$^2$) in the Sharon region overlying the Israeli coastal aquifer in which the abovementioned phenomena are observed was selected to investigate the process through calibrated flow and nitrate transport models from the agricultural land surface to the well screens (15 to 130 m below the surface). Unsaturated flow and nitrogen species transport models were calibrated to data from below the root zone that were obtained with direct push sampling under four typical crops in the area: citrus, persimmon, potato and strawberry. The flow and nitrate transport model in the aquifer was fed from water and nitrate fluxes from the unsaturated models, and calibrated to water levels and nitrate concentrations in the wells. The agricultural data and the flow and transport models of the unsaturated zone successfully predicted the total mass of nitrate in the aquifer. However, they failed to predict the spatial variability of nitrate in the wells, which was observed to be significantly larger than predicted. Therefore, the solution for calibrating the nitrate transport model was to multiply the modeled nitrate fluxes at the water table in small areas around the most contaminated wells with high multipliers (2.8–10), whereas nitrate fluxes in larger areas around the non-contaminated wells were multiplied by low factors (0.1–0.6) and in most of the area (55 %), the modeled fluxes from the unsaturated zone were conserved. The calibrated flow and transport model was then used to predict the development of nitrate concentrations in the aquifer 40 years in the future, with three nitrate-fertilization scenarios: business as usual (continuing present practice), or reducing nitrogen inputs by 25 % or 50 %. None of the scenarios showed any improvement in aquifer conditions in the next 10 years. Reducing nitrate application by 50 % will bring the average nitrate concentration in the aquifer to below drinking water standards in 40 years, whereas a cut of 25 % will only bring it back to the current level in 40 years. We conclude that the total mass of nitrate in an aquifer under agricultural land can be calculated with significant success from relatively limited land-use and deep unsaturated-zone data. Nevertheless, highly contaminated wells, are most probably effected by malfunction in the close vicinity of the well that cannot be predicted by a straight-forward agro-hydrological modeling scheme. Locally, it was shown that remediation of the aquifer in a half-century time scale requires reduction of the nitrogen fertilization input in the range of 25 % -50 %.

*Competing interests.* The authors declare that they have no conflict of interest.

*Acknowledgements.* The research leading to these results received funding from the Israeli Water Authority under research contract number 4500571791 as well as from the Chief Scientist, Ministry of Agriculture under contract numbers 304-0431-09 and 20-13-2013. Research was performed at the Agricultural Research Organization, Volcani Center.



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
