# Peer review of "Modeling nitrate from land-surface to wells' perforations under agricultural land: success, failure, and future scenarios in a Mediterranean case study."

_Hydrology and Earth System Sciences, 2017_

## Referee Comment (RC1) · Anonymous Referee #1 · 30 Mar 2017

I enjoyed reading this manuscript. The topic of nitrate variability in groundwater is important for environmental concerns and raises scientific questions about causes of spatial gradients. The meaning of the text is generally clear. The data and model results are interesting. I recommend moderate revisions to clarify some details of the models and to further emphasize the scientific implications of the work.

Title – consider revising to emphasize the main scientific issue (spatial variability of nitrate?). One option is to replace "success, failure" (which can be misunderstood) with "spatial variability." Abstract - Consider stating the scientific problem early in the

abstract (e.g. Can spatial variability of nitrate, be characterized on the basis of land use and standard agricultural practices?)

61 – Consider also mentioning that nitrate is discharged to streams or other surface water receptors, which can be a major concern.

64 - should this say "significant spatial variability"?

75-80 – In the statement of objectives, consider making the scientific implications (e.g. explaining the spatial variability of nitrate) more prominent, and perhaps de-emphasize the model-specific and site-specific elements.

77 - should "restore" be "estimate"?

100 – consider defining aerial coefficient of variation mathematically

145-150 – Are agricultural-chemical source of Cl important (e.g. KCl)? Are these accounted for in the mass balance?

227 - consider spelling out "Israel Water Authority" here.

245 - consider changing "strictly kept" to "kept constant" or something similar

248 - Section 2.3.3. – This is quite brief and readers will have additional questions, e.g. about initial conditions and boundary conditions for NO3– concentrations.

297 – Table 3 – Spell out "Crop Mass Balance" or define CMB in caption or table footnote.

306 – spell out MAE (mean absolute error?)

307 - It is not clear what is meant by "the improvement in the calibration ceased when...". Is the meaning that calibration efforts were stopped when MAE

316 - consider revising to "mean nitrate concentration for the entire modeled area"

318-319 – "The model reconstructed..." This seems repetitive and can be omitted.

326 and onward – It seems that the need for "multipliers" is a key result of the paper, because it indicates that nitrate variability is greater than can be explained by variation of crop-specific agricultural practices and physical processes, to the extent that they are simulated here. I suggest revising to emphasize this scientific significance, and to put less emphasis on the technical role of multipliers as an ad-hoc solution to a modeling problem. In other words, consider revising the language so that readers can see that the two models (with and without multipliers) address the scientific question of whether nitrate variability can be explained by general crop-type practices and the other factors considered in the numerical models.

Also, it would be helpful to further emphasize in the discussion how this result fits into the existing literature. For example, homogeneous NO3- input functions have been used with some success in local-scale (e.g., single field) Liao et al., studies to explain spatially varying NO3- concentrations (e.g. 2012 http://onlinelibrary.wiley.com/doi/10.1029/2011WR011008/full ; Alikhani et al.. 2016 http://www.sciencedirect.com/science/article/pii/S0022169416302098). In regional scale studies, it has been established that a homogeneous input function typically does not suffice, and multipliers similar to those of this study have been implemented (e.g. Green et al., 2016 http://www.sciencedirect.com/science/article/pii/S0022169416302852). This current study can be seen as a logical extension of the previous studies because it tests the extent to which the input function of NO3- can be improved, or even directly estimated from general agricultural practices and vadose zone characteristics. So in combining the current and previous studies, perhaps the authors could comment further on typical scales of variability (e.g. if intra-field variability of fertilizer applications were an issue, would the previous field-scale studies with homogeneous N-inputs have succeeded as well as they did?), the factors that may account for the variability

HESSD
(some already included in discussion), and/or related topics to inspire future research directions.

335 – consider adding a sentence to note that the physical significance of the multipliers will be addressed in the discussion section.

349 – change "on average for" to "as an average of". Consider clarifying/acknowledging that even though the average is less than 70 mg/L, there would still be some wells exceeding that limit.

362 - consider changing "coarse" to "approximate" or "first-order" or something similar

368 – I suggest not using the word "failure", as it can be misinterpreted as referring to the model itself, rather than to the relative smoothness of NO3– spatial gradients in the model as compared to measurements – a result which successfully addresses the scientific objectives of the study.

370 – The specific explanation here (intra-field variability) seems to be given without consideration of additional possibilities that are discussed later in this section (E.g. rapid transport in bore hole annulus)

376-378 – I don't follow the logic of this text.

378 - I suggest changing "non-physical" to "heuristic".

**HESSD**

---

## Referee Comment (RC2) · Anonymous Referee #2 · 26 Apr 2017

The manuscript is interesting and deals with a topic of great relevance around the world and some ideas are promising. However, the methodology seems to me a Little deficient so the results and the conclusions are compromised. Some specific comments are listed below, indicating the letter 'L' the line in the original manuscript: L35-45: The sentences in this paragraph are true all of them but all of them are really strong statements and sometimes a little bite unconnected between them. L46-50: Even when it is true, different crops and different regions present different efficiencies. I recommend you to try to explain this variability but also conclude with some results for Israel (or other semiarid regions) with some management more similar to your region (trees,

vegetables (not in green houses),...) L51: urea is mainly considered a synthetic fertilizer. L53: the importance of mass transport process is usually referred to NO3, but NH4 uptake usually occurs by diffusion. L55: consider changing light soils by aerated, dry,... L55: define 'relatively thin', because you are considering in your calibration at least 45 cm, and it could be even deeper. L58: denitrification to N oxides could be negligible in aerated soils, but complete denitrification to N2 is not so negligible and it is very difficult to measure, so there are a lack of real data. L59: this sentence is partially true, because nitrate leaching is not only the result of the nitrification, it is also the result of the poorly fixation of the nitrate molecule (negative charge) to the soil complex, mostly dominated by negative charges (clay and organic matter), whereas the ammonium (positive charge) presents a stronger retention to the soil and leaching is more difficult. L83-86: These two sentences fit better in the introduction. L86: if it is unconfined, do you know how much water and nitrate leave the system? It is important in order to predict if the new entries are greater or smaller. L100: Do you know if all the wells are extracting at the same depth? In some aquifers has been reported nitrate stratification, suggesting contamination from different time periods. L118-120: how is the irrigation applied? Is not the same if is homogeneous (surface, furrow, sprinkle) or if it is drip irrigation. L119: and what happen with the citrus? L120: please, define the size of the plastic tunnels, because there are many kinds. Moreover, the rainfall over the plastic should go somewhere, perhaps is draining with a reduced amount of nitrate, so this could lead to a reduction of the nitrate concentration in the aquifer. Please, discuss or consider it. L125: was it the same fertilizer rate when the petrol was cheaper some years ago? N fertilizers use to be highly related to energy price. L128: I do not understand why each crop should be correlated to each soil type. To me it makes more sense to have a soil map, combined with a weather map (you are presenting different precipitations) and with a nitrate concentration in the irrigation water map, and all them combined with the cop map, resulting in multiple combinations. Perhaps some of the variation that you can not explain is due to your simplification. L128: Have you try to make first a comparison of the observed concentration at each well with the percentage

of each crop in the well proximity? Because if it is not related, the rest of the assumptions could be not true. L129-138: Please define better the process. If I understood well you obtained for each crop three cores per depth; but, how do you divide them in order to get samples dried at 105°C and at 40°C at the same time. Moreover, do you think that if you dry the sample during 3 days at 40°C the soil nitrate and ammonium is going to be the same than analysed in fresh samples conserved in the refrigerator few hours/days? And mostly in deeper layers, because as you say mineralization, nitrification, denitrification and all the N processes could be enhanced by this temperature increase, doesn't they? Please discuss or define better. L184: I understand modelling simplification, but if there are some farms close to the region (as you propose as cause of the well differences) I have some doubts about the application of only this kind of compost. Could you discuss a little bit. L189: I understand that you try to fix your simulated data after 50 years to the values observed in the 12 cores; however, what are your initial data? Moreover, you said that the same crops have been cultivated during 15 years but you simulate 50. Is the same water table now than 50 years ago? (Probably with smaller irrigated area). Could you discuss that? L193: Can you define better what slightly means? L200-202: And what do you expect to happen with the N movement? Because in the no crop plots there also mineralization and nitrate leaching. L208: calibration is a main part in a modelling process, but independent validation also is. Because you calibration can sometimes be tricky, because you have many ways to get to a good result if you are combining many different parameters, but not only one of this ways is the more accurate. Because of that I suggest you to divide some observed results (in time, space or whatever) and use them for a new calibration and confirm that (validate) simulating for the other points and getting also an accurate adjust to the observed results. L234: Again, why are you using now 20 years instead of the 15 that are sure with the same management? L234-235: Consider including a reference for this statement. L237: the figure labels for each zone (particularly Bnei-Zion) are oversized and do not allow to see at least one sampling point and the scale. L238-239: the figure caption for b) and c) are changed. L242: please, define the period

observed for wells. L250-252: How do you expect that this change affect the crops? Do you have a crop module? And some of these data fit better in the results than here. L256: are they similar or are you using the same data? Define. L274: Why do you think that nitrate flux is 540 kg ha in the persimmon A if you only apply 200 kg? L315: consider including units. L330-332: how do you define which region should be multiplied by which factor? I do not understand this arbitrary correction. Table 5: define which coefficient for each one, please. L343-345: this has been already defined in the material and methods L369-370: It could be many other things, the simplification level of the system, the soil variability (you only sampled 12 cores for a 13.3 km2 surface), different soil/rainfall/management/nitrate in the well irrigation water/... for the same crop. L370-373: if you do not completely trust your data; how do you expect that we could do? L379: I could agree with you, but you should validate these coefficient in order to see that they are not just a mathematical trick. Figure 10: check the image quality. L417: I do not think that this is a "significant success" without some kind of validation

---

## Author Comment (AC1) · 18 May 2017

Authors' response to Referee #1 Comment

We thank the anonymous referee for his thoughtful review. We are sure the changes that will be made due to these comments will improve the manuscript significantly. The referee's comments and our responses are listed herein one by one.

1) Title – consider revising to emphasize the main scientific issue (spatial variability of nitrate?). One option is to replace "success, failure" (which can be misunderstood) with "spatial variability."

[Figure]

Response 1 - We considered this comment for a long time (as well as before submission). The scientific issue in this work is not only spatial variability but also how can we model the fate of nitrogen from the agricultural fields to nitrate contamination in groundwater wells. Putting spatial variability in the title can raise an expectation that the paper focuses on heterogeneity and finds solutions in the stochastic-hydrology arena. Therefore we prefer not to use this term in the title. Nevertheless, we took the reviewers suggestion to put spatial variability up front at the beginning of the abstract (comment # 2). The terms success and failure may be misunderstood, yet, on the other hand they raise positive curiosity in the reader. Reading the abstract is enough to understand the title. Furthermore, we think the success in modeling large-scale while failing in point estimation and the reasons for that in this case, is a significant part of the scientific conclusions of this study.

2) Abstract - Consider stating the scientific problem early in the abstract (e.g. Can spatial variability of nitrate, be characterized on the basis of land use and standard agricultural practices?)

Response 2 – The reviewer's suggestion was accepted the following will be added to the text in line 14 after the first sentence of the abstract: "Contaminated areas often show large spatial variability of nitrate concentration in wells. In this work we tried to assess whether this spatial variability, can be characterized on the basis of land use and standard agricultural practices."

3) 61 - Consider also mentioning that nitrate is discharged to streams or other surface water receptors, which can be a major concern.

Response 3 – The paper is about nitrate leaching to aquifers under Mediterranean climate in which permanent surface flow are rare. Nevertheless, we will add a sentence that mentions the ecological problem of accesses nitrogen from agricultural sources in surface water bodies.

4) 64 - Should this say "significant spatial variability"?

Response 4 - We thank the reviewer very much for this rightful correction. We will change "distribution" to "variability" in the revised manuscript.

5) 75-80 - In the statement of objectives, consider making the scientific implications (e.g. explaining the spatial variability of nitrate) more prominent, and perhaps de-emphasize the model-specific and site-specific elements.

Response 5 - As mentioned above the spatial variability is one aspect of the scientific implications in this paper but not the only one. The term spatial distribution will be changed here to spatial variability, which will make it more prominent.

6) - Should "restore" be "estimate"?

Response 6 – We will change the text to "reconstruct the observed groundwater nitrate concentrations". The term "reconstruct" is used often to describe a model results that fit observations.

7) 100 - Consider defining aerial coefficient of variation mathematically

Response 7: The sentence will be changed to: "The coefficient of variation (standard deviation / Average) of nitrate concentration in the wells in Fig. 2 is 38%."

8) 145-150 - Are agricultural-chemical source of Cl important (e.g. KCl)? Are these accounted for in the mass balance?

Response 8 - Usually potassium fertilizers in this area are added according to soil or leaf analysis and most farmers use mixed type K fertilizers when needed ($K_3PO_4$ , KCl, $K_2SO_4$ etc.). Therefore, we couldn't estimate the Cl concentration from fertilizer properly and it was neglected. In the worst case scenario of high need for K and using only KCl, the Cl mass contributed from the fertilizer will not exceed 15% of the total Cl mass input at the soil surface (Citrus orchard data, irrigation water contain 140 mg/l chloride and are the dominant source of Cl). This small contribution and rareness of such worst-case scenario justify neglecting this source of chloride in the chloride mass balance.

9) 227 - Consider spelling out "Israel Water Authority" here.

Response 9 - "IAW" will be changed to "Israel Water Authority".

10) 245 - Consider changing "strictly kept" to "kept constant" or something similar.

Response 10 - No, the recharge fluxes are not constant they are transient. They were calculated by the unsaturated zone flow models, and were not changed during calibration of the groundwater model.

11) 248 - Section 2.3.3. – This is quite brief and readers will have additional questions, e.g. about initial conditions and boundary conditions for NO3- concentrations.

Response 11- We accept the reviewers comment that this section is too brief and details on initial and boundary conditions are missing. Lines 256-258 will be changed to: "In the groundwater transport model the initial condition was the measured nitrate concentration at 2012. The transient nitrate-concentration boundary conditions were modified to account for similar reductions in nitrogen fertilization outside of the model domain. This was done by two steps: (1) run the model to the future with constant boundary condition and looking on the trends of the nitrate concentration of the wells inside the model domain; (2) adjusting these trends to the boundary condition and run the model to the future again with transient boundary conditions."

12) 297 - Table 3 – Spell out "Crop Mass Balance" or define CMB in caption or table footnote.

Response 12 –"Chloride Mass Balance" will be spelled out in the Table caption

13) 306 – spell out MAE (mean absolute error).

Response 13 – mean absolute error was defined as MAE in line 243.

14) 307 - It is not clear what is meant by "the improvement in the calibration ceased when. . .". Is the meaning that calibration efforts were stopped when MAE <0.5 and bias <0.1? And/or that it was difficult to improve results beyond those cutoff values?

Response 14 – The meaning is that the calibration efforts were stopped when the conditions of average MAE <0.5 m and bias <0.1 m, were achieved. It was assessed that in the framework of this research this target is appropriate. The hydraulic head gradient in the research area is relatively linear from east to west, with a magnitude of about 2.5 m/ 1 km, hence, the target of 0.5m is reasonable. We also do not want to elaborate further on the flow system in this paper because the main issue is the nitrogen transport and fate. Due to the reviewer's comment "met" in line 308 will be changed to "achieved"

15) 314 - Should this say "initial transport parameters"? (is this 500m value the one referred to in the previous sentence?)

Response 15 – No, this value of longitudinal dispersivity is the fitted value after the first stage of calibration. To emphasize this point we will change the text to: "The final transport parameters used in the calibrated model ..."

16) 316 – Consider revising to "mean nitrate concentration for the entire modeled area"

Response 16 - We thank the reviewer for this good suggestion: "nitrate concentrations over the entire modeled area" will be changed to: "mean nitrate concentration for the entire modeled area".

17) 318-319 - "The model reconstructed. . ." This seems repetitive and can be omitted.

Response 17 – We thank the reviewer for this comment. Nevertheless, although there is redundancy in this text we prefer to leave it because the first scentence describes the goodness of fit measures, while the second is the mechanistic interpretation of this result. The words "This means. . ." will be added to the beginning of the second sentence to emphasize the point.

18) 326 and onward – It seems that the need for "multipliers" is a key result of the paper, because it indicates that nitrate variability is greater than can be explained by variation of crop-specific agricultural practices and physical processes, to the extent that they

are simulated here. I suggest revising to emphasize this scientific significance, and to put less emphasis on the technical role of multipliers as an ad-hoc solution to a modeling problem. In other words, consider revising the language so that readers can see that the two models (with and without multipliers) address the scientific question of whether nitrate variability can be explained by general crop-type practices and the other factors considered in the numerical models. Also, it would be helpful to further emphasize in the discussion how this result fits into the existing literature. For example, homogeneous $NO_3^-$ input functions have been used with some success in local-scale (e.g.. single field) studies to explain spatially varying $NO_3^-$ concentrations (e.g. Liao et al., 2012 http://onlinelibrary.wiley.com/doi/10.1029/2011WR011008/full ; Alikhani et al., 2016 http://www.sciencedirect.com/science/article/pii/S0022169416302098). In regional scale studies, it has been established that a homogeneous input function typically does not suffice, and multipliers similar to those of this study have been implemented (e.g. Green et al., 2016 http://www.sciencedirect.com/science/article/pii/S0022169416302852). This current study can be seen as a logical extension of the previous studies because it tests the extent to which the input function of $NO_3^-$ can be improved, or even directly estimated from general agricultural practices and vadose zone characteristics. So in combining the current and previous studies, perhaps the authors could comment further on typical scales of variability (e.g. if intra-field variability of fertilizer applications were an issue, would the previous field-scale studies with homogeneous N-inputs have succeeded as well as they did?), the factors that may account for the variability (some already included in discussion), and/or related topics to inspire future research directions.

Response 18 – We thank the reviewer for this important comment. The significant conclusion concerning aquifer-nitrate spatial variability and land use spatial variability is written in lines 324-325 in the results section. We tried to separate results from discussion in this paper, nevertheless the reviewer's suggestion to further discus the scientific (and practical) meaning here is accepted for improving reading flow. In line 325 the following text will be added: "The meaning of this, is that nitrate spatial variability cannot be explained only by physical process of agricultural practice and land-use variability on surface. Other factors that are local and arbitrary, significantly affect nitrate concentration in some wells and therefore the measured spatial variability of nitrate in the aquifer. These factors were introduced into the numerical model as will be explained hereafter." As of the discussion, we accept the comment and will include more references in the revised version to better fit to existing literature, as suggested (e.g. Alikhani et al., 2016; Kourakos et al., 2012; Spalding and Exner., 1993; Liao et al., 2012; Green et al., 2016 etc.).

19) 335 – Consider adding a sentence to note that the physical significance of the multipliers will be addressed in the discussion section.

Response 19 – A sentence before, the reader is directed to the discussion section (line 333). The term "physical significance" will be used as suggested

20) 349 – Change "on average for" to "as an average of". Consider clarifying/acknowledging that even though the average is less than 70 mg/L, there would still be some wells exceeding that limit.

Response 20 – "on average for" will be changed to: "as an average of" as suggested. We thank the reviewer for the second remark. A sentence will be added saying: "Even in this case about half of the wells will still exceed the standard concentration."

21) 362 – Consider changing "coarse" to "approximate" or "first-order" or something similar

Response 21 – "Coarse" will be changed to "first-order"

22) 368 – I suggest not using the word "failure", as it can be misinterpreted as referring to the model itself, rather than to the relative smoothness of NO3- spatial gradients in the model as compared to measurements – a result which successfully addresses the scientific objectives of the study.

[Figure]

Response 22 - The strait-forward mechanistic modeling scheme that was used failed to reproduce the spatial variability in wells. Nevertheless, this result and modeling scheme led to the proof that unrealistic nitrate fluxes (much higher than application rates) are needed to support the most contaminated wells.

23) 370 – The specific explanation here (intra-field variability) seems to be given without consideration of additional possibilities that are discussed later in this section (E.g. rapid transport in bore hole annulus)

Response 23 - High fluxes of nitrate can be a consequence of high fluxes of water (hole annulus, leaks of irrigation system etc.) and/or high concentrations of nitrate (fertilizer tank leakage, compost pile forgotten to be distributed etc.). In line 369-370 we will add a clarification on possible reasons for high nitrate inputs close to the well: "...and are a result of random failure of even fertilizer distribution in the field that can be due to one or more of the following reasons."

24) 376-378 – I don't follow the logic of this text.

Response 24 – We understand the paragraph is somewhat unclear and will be clarified in the revised paper. The first sentence argues that heterogeneity of the porous medium may also cause local very high nitrate fluxes. The second sentence shows how data from this study supports such possibility. The text will be changed to the following: "Heterogeneity of the porous medium may cause extremely high nitrate fluxes likewise well failure discussed previously, and may be a source for local high contamination. The field survey reported here support this statement. Of the nine deep profiles reported here (Figure 6, Table 1), one showed extreme nitrate concentrations and calculated nitrate fluxes that were 4- to 5-fold higher than in the other profiles extracted from the same orchard (Persimmon A, Table 1). "

25) 378 – I suggest changing "non-physical" to "heuristic"

Response 25 - We accept the comment. "non-physical" will be changed to "heuristic"

---

## Author Comment (AC3)

**Authors' response to referee #2 comments**

We thank referee #2 for his constructive review. We are sure that changes made in the manuscript due, will improve it. The referee's comments and our responses are listed herein one by one.

1)     The manuscript is interesting and deals with a topic of great relevance around the world and some ideas are promising. However, the methodology seems to me a little deficient so the results and the conclusions are compromised. Some specific comments are listed below, indicating the letter 'L' the line in the original manuscript:

**Response 1:** Modeling of flow and transport from land surface to well- perforations (20 – 100 m deep, from which 5-50 are unsaturated) at an agricultural area with various crops of 13 $KM^2$, will always be deficient. Hence, model results and conclusions may be viewed as a compromise. We believe the data-based modeling in this work is a worthy and skill-full effort and the conclusions are significant, novel to some extent, and of high interest for the hydrology research and practice community.

**2) L35-45:** The sentences in this paragraph are true all of them but all of them are really strong statements and sometimes a little bite unconnected between them.

**Response 2**: The rational of this paragraph is to shortly describe the origin, a little history, and the magnitude of the global "nitrate problem" in groundwater with its relation to agriculture, hydrology and water quality. Following this comment we will slightly revise the last 3 sentences in the paragraph to increase the connection between the sentences, as follows: "Thus nitrate has become the most common groundwater contamination caused by agricultural activity in many countries (Jalali, 2005; Vitousek et al., 2009; Burow et al., 2010; Kourakos et al., 2012; Yue et al., 2014; Wheeler et al., 2015; Wang et al., 2016). In Israel for example, more than half of all the wells that have been disqualified as sources of drinking water were disqualified due to nitrate contamination (Israel Water Authority; IWA, 2015a). The process of groundwater contamination by nitrate occurs mainly below light soils and less under cultivated clays (Spalding and Exner, 1993; Kurtzman et al., 2016)."

**3)** **L46-50:** Even when it is true, different crops and different regions present different efficiencies. I recommend you to try to explain this variability but also conclude with some results for Israel (or other semiarid regions) with some management more similar to your region (trees, vegetables (not in green houses),. . .)

**Response 3**: We agree that different crops at different regions present different efficiencies, exactly because of that we mentioned a large range of nitrate leaching percentages and we wrote: "…in different crops and countries" and cited many studies from around the world. From Israel we cited a large research that was made over many years (35) for many crop types (18) and a more recent work in modern greenhouses. However, as of the comment we will add "(18 crop varieties)" after "vegetables and fields crop" (line 49)

**4)** **L51:** urea is mainly considered a synthetic fertilizer.

**Response 4**: Synthetic or natural, urea ($CO(NH_2)_2$) is an organic compound and as of processes in the soil (and model) it undergoes mineralization, like the other organic nitrogen forms.

**5)** **L53:** the importance of mass transport process is usually referred to NO3, but NH4 uptake usually occurs by diffusion.

**Response 5**: Mass transport include both advection and dispersion, in which molecular diffusion is dominant in low velocities. We will add "(advective and diffusive)" after process.

**6)** **L55:** consider changing light soils by aerated, dry,. . .

**Response 6**: The sentences will be corrected to: "…and in aerated light soil,"

**7)** **L55:** define 'relatively thin', because you are considering in your calibration at least 45 cm, and it could be even deeper.

**Response 7**: The nitrification occurs mainly at the upper part of the unsaturated zone, close to the land surface. Data of nitrification potential in orchards from this area

shows that most of the potential is in the top 15 cm and almost negligible below 45 cm.

0-45 cm as the layer of nitrification and reference to Kurtzman et al., 2013 will be added to the revised manuscript.

8) **L58:** denitrification to N oxides could be negligible in aerated soils, but complete denitrification to N2 is not so negligible and it is very difficult to measure, so there are a lack of real data.

**Response 8**: In this sentence we cited studies that neglected denitrification in their models. Denitrification is small but not negligible in the models of this work (Table 2, 3b). Denitrification is a sink for N-NO3 in our models and the further fate of the N species are not part of the scope of this work.

9) **L59:** this sentence is partially true, because nitrate leaching is not only the result of the nitrification, it is also the result of the poorly fixation of the nitrate molecule (negative charge) to the soil complex, mostly dominated by negative charges (clay and organic matter), whereas the ammonium (positive charge) presents a stronger retention to the soil and leaching is more difficult.

**Response 9**: We thank the reviewer for the good comment. The following sentence will be added to the revised manuscript: "Moreover, ammonium is a cation and tends to adsorb to the soil solids (clay fraction, organic matter)" (Line 56).

10) **L83-86:** These two sentences fit better in the introduction.

**Response 10:** We wrote these sentences in the methods (research area section) to explain the choice of research site. This makes the text flow better.

11) **L86:** if it is unconfined, do you know how much water and nitrate leave the system? It is important in order to predict if the new entries are greater or smaller.

**Response 11:** We mentioned the aquifer is unconfined because a major part of this work deals with water and nitrate fluxes entering the aquifer from the unsaturated

zone above. Water and nitrate enter/exit the modeled aquifer domain from the side boundaries and exits also through pumping wells.

The nitrate budget in the aquifer is always calculated, when concentration increase entries exceed exits and when concentrations decreases exits are higher than entries (e.g. scenario of reduced N fertilization).

12) **L100:** Do you know if all the wells are extracting at the same depth? In some aquifers has been reported nitrate stratification, suggesting contamination from different time periods.

**Response 12**: We show in Fig. 5(c) the depth of well-screens in the groundwater modeled area. Data does not show a trend of nitrate with screen depth. We discuss the possibility of denitrification in deep parts of the aquifer earlier in the text in lines (60-64) including a citation.

13) **L118-120:** how is the irrigation applied? Is not the same if is homogeneous (surface, furrow, sprinkle) or if it is drip irrigation.

**Response 13:** We will refer to the irrigation technique in the revised manuscript and text will be changed to: "The potato field was irrigated by sprinklers with an average…" "The strawberry field was irrigated by micro-sprinklers (at the early stage of growing) and drip irrigation after, to an average…"   "The persimmon orchard was irrigated by micro-sprinklers to an average…"

14) **L119:** and what happen with the citrus?

**Response 14**: As mentioned a few lines before (line 115), the data and unsaturated flow and transport model for citrus were taken from Kurtzman et al. (2013).

15) **L120:** please, define the size of the plastic tunnels, because there are many kinds. Moreover, the rainfall over the plastic should go somewhere, perhaps is draining with a reduced amount of nitrate, so this could lead to a reduction of the nitrate concentration in the aquifer. Please, discuss or consider it.

**Response 15**: We do not wish to overload the manuscript with more agro-technical data, which does not contribute to the main theme (nitrogen fate from ground surface to wells). After the tunnels are set irrigation is by drippers within the tunnel, and rain drains on top of the continuous plastic sheets out of the field, hence, it has no effect on the transport of nitrogen below the field.

16) **L125:** was it the same fertilizer rate when the petrol was cheaper some years ago? N fertilizers use to be highly related to energy price.

**Response 16**: No. As mentioned before, our knowledge about the agricultural practice is based on the farmers reports concerning the 10-20 years prior to soil sampling (where needed, we used the extension-service recommendations). There was no mentions of changes in the rate of N fertilization due to its price by the farmers.

17) **L128:** I do not understand why each crop should be correlated to each soil type. To me it makes more sense to have a soil map, combined with a weather map (you are presenting different precipitations) and with a nitrate concentration in the irrigation water map, and all them combined with the cop map, resulting in multiple combinations. Perhaps some of the variation that you cannot explain is due to your simplification.

**Response 17:** The soil-map which represent the soil type at surface will show the same soil for the entire modeled area as the first order classification – Red Mediterranean relatively sandy - "Hamra" soil (sandy loam - loamy sand up to sandy texture in some places). Therefore, in the root zone the sediment is quiet homogenous, yet heterogeneity in deep profiles is much larger. The 2 main limitations from having a calibrated model for the unsaturated zone for each plot are: 1) deep sampling cost and availability; 2) time and skills needed for model calibration. Therefore, for upscaling from only one field that was sampled (per land-use) to the large region, we decided to simplify by using one calibrated unsaturated zone model (which was extended to dozens of models for each unsaturated-zone thickness) for each crop type. This methodology is described in lines 157-158 and 186-202, and the simplifications are discussed in lines 361-378.

**18) L128:** Have you try to make first a comparison of the observed concentration at each well with the percentage of each crop in the well proximity? Because if it is not related, the rest of the assumptions could be not true.

**Response 18:** We didn't make this comparison because we aim at a much higher target than showing a 2D relationship between nitrate concentration in wells and land use which maybe misleading (unsaturated-zone, perforation depths and pumping rates are variable, there is a natural gradient, etc.,). We aim at describing the 3D flow and transport phenomenon from field surface to well perforations mechanistically, accounting for the above issues (see lines 75-80).

**19) L129-138:** Please define better the process. If I understood well you obtained for each crop three cores per depth; but, how do you divide them in order to get samples dried at 105°C and at 40°C at the same time. Moreover, do you think that if you dry the sample during 3 days at 40°C the soil nitrate and ammonium is going to be the same than analysed in fresh samples conserved in the refrigerator few hours/days? And mostly in deeper layers, because as you say mineralization, nitrification, denitrification and all the N processes could be enhanced by this temperature increase, doesn't they? Please discuss or define better.

**Response 19:** As written in lines 129-130, for each crop 3 continuous cores from land surface to depth of 10 m depth were cored with the direct push technique. The cores were then cut into 30 cm segments and the segments were analyzed in the laboratory. From each segment a sample was taken for the gravimetric water content analysis (drying at 105 °C ) and the rest of the soil in the segment was taken to drying in 40 °C, grinding ,sieving at 2 mm and extractions for the chemical analysis. Drying is essential for grinding and sieving the soil and getting a representative and extractable sample before extraction, (we will insert "grind and" before "sieved" in line 137).

We used the same method here as performed in the work on citrus orchards which we use some of its results in the current analysis (Kurtzman et al. 2013). Prior to that work tests were made comparing extraction from fresh – dry – dry + grined and sieved samples and the average results of nitrate and ammonium were mostly the same (or

lower in the fresh samples due to less effective extraction). The main problem with fresh samples was the large differences between replicates from the same core-segment, relative to the good homogenization of the sample after drying and sieving.

The choice of drying at 40 °C rather than warmer or cooler is to balance among the necessity of drying, the impact on the concentrations (mainly due to ammonium volatilization) and the drying duration. Drying soil in 40 °C for nitrate analysis is a common practice in soil science (e.g. Bottomley, P.S., Angle J.S., and Weaver. R.W.: Methods of Soil Analysis: Part 2—Microbiological and Biochemical Properties SSSA Book Ser. 5.2., SSSA, Madison, WI., 1994.; Unkovich, M., Herridge, D., Peoples, M., Cadisch, G., Boddey, R., Giller, K., Alves, B. and Chalk, P.: Measuring plant-associated nitrogen fixation in agricultural systems, Australian Centre for International Agricultural Research (ACIAR), Canberra, Australia, 2008).

20) **L184:** I understand modelling simplification, but if there are some farms close to the region (as you propose as cause of the well differences) I have some doubts about the application of only this kind of compost. Could you discuss a little bit.

**Response 20:** In the referenced study in line 184 (Ben Hagai et al., 2011) many types of composts from different compost producers were analyzed and averages of composts physical and chemical properties are published. We use these average numbers for modeling nitrogen inputs due to compost application in our regional analysis. Of course this may be another source of variability, nevertheless we believe compost that is distributed well in the field will not cause major differences in nitrate concentration in wells in regional scale.

21) **L189:** I understand that you try to fix your simulated data after 50 years to the values observed in the 12 cores; however, what are your initial data? Moreover, you said that the same crops have been cultivated during 15 years but you simulate 50.
Is the same water table now than 50 years ago? (probably with smaller irrigated area). Could you discuss that?

**Response 21:** The idea of going way back to the past with the unsaturated-zone models is that the initial conditions will not play a significant role in the water and nitrate fluxes that feed the groundwater model starting in 1992 (30 years are sufficient for practice on the ground to reach the water table). Therefore, the actual initial conditions in the unsaturated zone assigned for 1962 are not important. Nevertheless we used reasonable conditions knowing the N concentrations - profile in 2012 and assuming the profile was poorer with N, 50 years before.

Orchards (citrus, persimmon) go back for a few decades in most plots, for the annuals (vegetable), of course more changes were made, in the surface inputs during the years. We took plots that farmers reported their agricultural practice for 15 years because our calibration data (10 m depth) is influenced from this period.  This work did not aim at collecting exact history of cultivation for the last 50 years and a steady agro-practice was assumed. We will add in line 190 "… under the assumption of steady crop and agricultural practice during the 50 years"

As for the water-table, its depth is relatively stable between 1992-2012 therefore these years were chosen as the modeling period of the groundwater model (see figure in the file attached to this response). Doing so the fluxes from the unsaturated zone models were taken as the fluxes at a fixed depth (no water-table influence). Before 1992 the water-table was at lower elevation due to more pumping in this region (see figure in the file attached to this response), data from the Israel water authority). The aforementioned details are in the manuscript in lines 231-235, and we believe further elaboration on this issue will not contribute to the manuscript.

[Figure]

**22) L193:** Can you define better what slightly means?

**Response 22:** Slightly means changes in the saturated hydraulic conductivity were in the same order of magnitude. This means the initial hydraulic properties suggested by the pedotransfer functions were not changed dramatically during calibration.

In line 193 after "slightly changes" we will add: "…(i.e. only Ks within the same order of magnitude)…"

**23) L200-202:** And what do you expect to happen with the N movement? Because in the no crop plots there also mineralization and nitrate leaching.

**Response 23:** We thank the referee for this good comment. Small surface fluxes of nitrogen exist in the non -cultivated areas. In the no-crop model we applied 10 kg ha$^{-1}$ yr$^{-1}$ nitrogen with some spreading during the year.

In line 201 we will add after "water flow": "and nitrogen transport (10 kg ha$^{-1}$ yr$^{-1}$ nitrogen applied on ground surface)…"

**24) L208:** calibration is a main part in a modelling process, but independent validation also is. Because you calibration can sometimes be tricky, because you have many ways to get to a good result if you are combining many

different parameters, but not only one of this ways is the more accurate. Because of that I suggest you to divide some observed results (in time, space or whatever) and use them for a new calibration and confirm that (validate) simulating for the other points and getting also an accurate adjust to the observed results.

**Response 24:** The workflow in this work was tight after sampling and analyzing, 3 unsaturated-zone models were set and each one calibrated for flow than the transport model for each flow model was set and calibrated (2 stages, dispersity with chloride data, and than the reactive transport of nitrogen). Following that the saturated 3D groundwater flow model was calibrated and than the nitrate transport in the aquifer. Adding a validation stage for each calibration would have given a statistical measure of the goodness of fit of the calibrated model with independent observations for each stage. Goodness of fit is not an objective of this work, therefore we do not think the validation process suggested by the referee would have gained us better understanding of what can and cannot be done in this type of modeling.

Generally speaking, modern modeling does not look as highly as in the past on validation. e.g. Anderson et al. (2015) Applied Groundwater Modeling P. 19 "The terms model verification, code verification, and model validation are not in the workflow because verification and validation, as historically used, are no longer critical elements in groundwater modeling"

25) **L234:** Again, why are you using now 20 years instead of the 15 that are sure with the same management?

**Response 25:** The 15 years which the farmers in the sampled plots reported the agricultural practice were needed for the calibration of the 10 m deep unsaturated zone models. Moving to the regional analysis where the unsaturated-zone models were extended/shortened in depth for each cell of the saturated zone model – the 15 years reported by the specific farmers of the sampled plots, has no meaning any more. The reason for modeling the aquifer for the years 1992-2012 is explained in lines 231-235 and further here in response 21.

**26) L234-235:** Consider including a reference for this statement.

**Response 26:** in line 235 after "stable" we will add "(Israel Water Authority data),…"

**27) L237:** the figure labels for each zone (particularly Bnei-Zion) are oversized and do not allow to see at least one sampling point and the scale.

**Response 27:** We thank the reviewer for this comment the graphics of Figure 5 will be changed in a way that no spatial data will be hidden.

**28) L238-239:** the figure caption for b) and c) are changed.

**Response 28:** Oops, the caption numbering (b,c) in Figure 5 will be corrected, thanks.

**29) L242:** please, define the period observed for wells.

**Response 29**: in line 242 after "in the wells" we will add "(1992-2012)".

**30) L250-252:** How do you expect that this change affect the crops? Do you have a crop module? And some of these data fit better in the results than here.

**Response 30**: This estimation is based on the calculations of Kurtzman et al. (2013) which is cited. Both agricultural and hydrological consequences were discussed in that study, whereas, in the current study we concentrate on the hydrological. Kurtzman et al. 2013 emphasize the point, that the reduction in N-root-uptake is smaller than the reduction in nitrate-leaching when N input is reduced from its high rates, because of the root-up-take data-supported model that is used there for citrus and in the same form (different parameters) here for vegetables and deciduous. From Kurtzman et al., 2013: a decrease of 25 % in the nitrogen fertilization mass results in a decrease of only 4 % - 9 % of root nitrogen uptake yet it resulted in a decrease of 50 % nitrate-nitrogen flux at the water table. Decrease of 50 % in the nitrogen fertilization mass results in a decrease of 22 % of root nitrogen uptake and a decrease of 72 % nitrate-nitrogen flux at the water table.

This short paragraph is included in the Materials and methods section because it explains the method used for the scenario simulations in this work. The results reported here are from Kurtzman et al. (2013) and not a results of this work.

**31) L256:** are they similar or are you using the same data? Define.

**Response 31**: Thanks for this comment. The same data was used. We will change the wording in the revised manuscript so it will be clear that the same data was used for future atmospheric boundary conditions.

**32) L274:** Why do you think that nitrate flux is 540 kg ha in the persimmon A if you only apply 200 kg?

**Response 32**: The N-NO3 deep fluxes reported in the bottom line of Table 1 are those calculated for each sampled profile using the Cl and N-NO3 concentrations of the deep sediment (Eq. 1 & 2). In this profile (persimmon A) the deep samples contained very high N-NO3 concentrations resulting in high calculate N-NO3 flux. As the referee commented rightfully, this flux is almost 3 folds higher than the surface N input the farmer apply on average on the orchard surface. This is a good example of the heterogeneity of N deep fluxes that is needed to support the concentrations in the contaminated wells, and it is discussed later (lines 375 – 377). If the N-NO3 fluxes to the water table were everywhere 20-50% of the applied-N rate (crop dependent) as they are on average, the concentrations of NO3 in wells was much more uniform than observed – this is a major result of this study.

**33) L315:** consider including units.

**Response 33**: the porosity and the dispersivities-ratio are dimensionless.

**34) L330-332:** how do you define which region should be multiplied by which factor? I do not understand this arbitrary correction.

**Response 34**: After the first round of calibration of the NO3 transport model we knew we get good the total mass of nitrate that enters the aquifer from the unsaturated zone. We also knew these fluxes are too uniform to produce the variability in nitrate concentrations in wells. We still assumed the variability in wells can only be caused by variable top fluxes. Therefore, we first asked in how much should we multiply the nitrate fluxes to get the high concentrations that we got in the most contaminated wells ([NO3]>100 mg/l). We found we need multipliers of 5 and even 10 for the fluxes

near these wells. To keep the total flux in the region the same, the small areas that got high multipliers had to be compensated with relatively large areas with low multipliers – these were naturally distributed around wells were the nitrate concentrations are relatively low. Most of the calculated fluxes from the field profiles (Table 1) were not very far from the average flux of our representative transient model for each crop (Table 3), hence there is good reasoning for keeping most of the area multiplied by 1. The range of 2 orders of magnitude for the multipliers was not chosen arbitrary it was shown in the work of Kourakus et al 2012 that nitrate fluxes cover 2 order of magnitude, this will be clarified in the revised text.

**35) Table 5:** define which coefficient for each one, please.

**Response 35:** The comment is not clear, what coefficient? what one?

If the referee meant what multiplier was assigned to each well in the 2$^{nd}$ calibration, the answer is that there is no one multiplier for each well. There are different multipliers for different model cells around each well. Generally, the high multipliers were assigned close to the most contaminated wells.

**36) L343-345:** this has been already defined in the material and methods

**Response 36**: That is correct. Nevertheless, we prefer to repeat shortly here (2 lines) for the reading -flow of the results section, and for readers who are only interested in the scenario simulations results.

**37) L369-370:** It could be many other things, the simplification level of the system, the soil variability (you only sampled 12 cores for a 13.3 km2 surface), different soil/rainfall/management/nitrate in the well irrigation water/. . . for the same crop.

**Response 37:** We generally agree and thank the referee for this comment and will change the wording here to be less strict. Yet, it should be emphasized that fluxes of nitrate that are a few folds larger than any reasonable fertilization rate (like the referee noticed in comment # 32), are needed to reconstruct the contamination in wells. This conclusion should be said somehow loud. The following discussion explains

(and will explain better in the revised text) possible reasons for uneven distribution of N by close-to-well high loads. Further, heterogeneity of porous-medium is mentioned as possible cause for variability in concentrations in groundwater. See also response to comment # 24 of Referee # 1 and the changes we suggested there.

**38) L370-373:** if you do not completely trust your data; how do you expect that we could do?

**Response 38:** We completely trust the nitrate-concentration of water sampled from wells and analyzed by the Israel Water Authority or authorized organizations and are reported in their database. These are the concentration in wells' water; the lines mentioned by the referee discus, what may lead to high concentrations in some wells and has nothing to do with the goodness of this data.

We also completely trust farmers report on level of fertilization in the following manner. For example, if a farmer uses liquid NH4NO3 as N-fertigation in his drip-irrigation system. We are positive he calculates well the mass of N he brings to the field in the fertilizer tank considering the irrigated area, and his irrigation/fertigation schedule to report X kg N ha-1 y-1 is applied (our models, of daily resolution, breakdown this with additional data from the farmer, to N applied at each irrigation). What we say in these lines is that in the vicinity of some wells, the fertilizer for some time, did not find its way to evenly be distributed in the area and higher fluxes occurred near the contaminated wells.

**39) L379:** I could agree with you, but you should validate these coefficients in order to see that they are not just a mathematical trick.

**Response 39:** If the referee means by "mathematical trick" that this specific set of coefficients and their distribution are not a unique solution for fitting the observed and modeled nitrate concentration, we partly agree. The # of multipliers, their values, and their spatial distribution may be further optimized, yet the optimized scheme will lead to the same important conclusions: 1) The most contaminated wells require local fluxes of nitrate that cannot be explained by regular agricultural practice and representative physically-based modeling ; 2) In most of the agricultural area the

fluxes produsced by the representative data-based models are a good estimate of the local nitrate fluxes at the water table;

As of the validation see response to comment # 24 here.

40) **Figure 10:** check the image quality.

**Response 40:** We want to preserve the original figure from Mercado 1976 (we believe the "old looking" gives the reader a good feeling of the post audit analysis 40+ year old predictions).

41) **L417:** I do not think that this is a "significant success" without some kind of validation.

Response 41: Here there is no question of validation. The unsaturated zone analysis and models gave a very good estimate of the total mass of nitrate that entered the aquifer from above, without using any data from the aquifer for calibration of the unsaturated-zone models.